# STRUCTURE AND RANDOMNESS IN PLANNING AND REINFORCEMENT LEARNING

## ABSTRACT

Planning in large state spaces inevitably needs to balance depth and breadth of the search. It has a crucial impact on planners performance and most manage this interplay implicitly. We present a novel method *Shoot Tree Search (STS)*, which makes it possible to control this trade-off more explicitly. Our algorithm can be understood as an interpolation between two celebrated search mechanisms: MCTS and random shooting. It also lets the user control the bias-variance trade-off, akin to $TD(n)$, but in the tree search context.

In experiments on challenging domains, we show that STS can get the best of both worlds consistently achieving higher scores.

## 1 INTRODUCTION

Classically, reinforcement learning is split into model-free and model-based methods. Each of these approaches has its strengths and weaknesses: the former often achieves state-of-the-art performance, while the latter holds the promise of better sample efficiency and adaptability to new situations. Interestingly, in both paradigms, there exists a non-trivial interplay between structure and randomness. In the model-free approach, Temporal Difference (TD) prediction leverages the structure of function approximators, while Monte Carlo (MC) prediction relies on random rollouts.

Model-based methods often employ planning, which counterfactually evaluates future scenarios. The design of a planner can lean either towards randomness, with random rollouts used for state evaluation (e.g. random shooting), or towards structure, where a data-structure, typically a tree or a graph, forms a backbone of the search, e.g. Monte Carlo Tree Search (MCTS). Planning is a powerful concept and an important policy improvement mechanism. However, in many interesting problems, the search state space is prohibitively large and cannot be exhaustively explored. Consequently, it is critical to balance the depth and breadth of the search in order to stay within a feasible computational budget. This dilemma is ubiquitous, though often not explicit.

The aim of our work is twofold. First, we present a novel method: Shoot Tree Search (STS). The development of the algorithm was motivated by the aforementioned observations concerning structure, randomness, and dilemma between breadth and depth of the search. It lets the user control the depth and breadth of the search more explicitly and can be viewed as a bias-variance control method. STS itself can be understood as an interpolation between MCTS and random shooting. We show experimentally that, on a diverse set of environments, STS can get the best of both worlds. We also provide some toy environments, to get an insight into why STS can be expected to perform well. The critical element of STS, *multi-step expansion*, can be easily implemented on top of many algorithms from the MCTS family. As such, it can be viewed as one of the extensions in the MCTS toolbox.

The second aim of the paper is to analyze various improvements to planning algorithms and test them experimentally. This, we believe, is of interest in its own right. The testing was performed on the Sokoban and Google Research Football environments. Sokoban is a challenging combinatorial puzzle proposed to be a testbed for planning methods by Racanière et al. (2017). Google Research Football is an advanced, physics-based simulator of football, introduced recently in Kurach et al. (2019). It has been designed to offer a diverse set of challenges for testing reinforcement learning algorithms.

The rest of the paper is organized as follows. In the next section, we discuss the background and related works. Further, we present details of our method. Section 4 is devoted to experimental results.

## 2    BACKGROUND AND RELATED WORK

The introduction to reinforcement learning can be found in Sutton & Barto (2018). In contemporary research, the line between model-free and model-based methods is often blurred. An early example is Guo et al. (2014), where MCTS plays the role of an 'expert' in DAgger (Ross & Bagnell (2014)), a policy learning algorithm. In the series of papers Silver et al. (2017; 2018), culminating in AlphaZero, the authors developed a system combining elements of model-based and model-free methods that master the game of Go (and others). Similar ideas were also studied in Anthony et al. (2017). In Miłoś et al. (2019), planning and model-free learning were brought together to solve combinatorial environments. Schrittwieser (2019) successfully integrated model learning with planning in the latent space. A recent paper, Hamrick et al. (2020), suggests further integration model-free and model-based methods via utilizing internal planner information to calculate more accurate estimates of the $Q$-function. Soemers et al. (2016) presents expansion much similar to ours. The crucial algorithmic difference is the aggregate backprop (for details see Section 3). In a similar vain, Coulom (2006), proposes a framework blending tree search and Monte-Carlo simulations in a smooth way. Both, Soemers et al. (2016) and Coulom (2006) differ from our work as they do not used learned value functions, resorting to heuristics and/or long rollouts. In James et al. (2017) a detailed empirical analysis suggests that the key to the UCT effectiveness is the correct ordering of actions. As most of these works, we use the model-based reinforcement learning paradigm, in which the agent has access to a true model of the environment.

Searching and planning algorithms are deeply rooted in classical computer science and classical AI, see e.g. Cormen et al. (2009) and Russell & Norvig (2002). Traditional heuristic algorithms such as A$^*$ (Hart et al. (1968)) or GBFS (Doran & Michie (1966)) are widely used. The Monte Carlo Tree Search algorithm, which combines heuristic search with learning, led to breakthroughs in the field, see Browne et al. (2012) for an extensive survey. Similarly, Orseau et al. (2018) bases on the classical BFS to build a heuristic search mechanism with theoretical guarantees. In Agostinelli et al. (2019) the authors utilise the value-function to improve upon the A$^*$ algorithm and solve Rubik's cube.

Monte Carlo rollouts are known to be a useful way of approximating the value of a state-action pair Abramson (1990). Approaches in which the actions of a rollout are uniformly sampled are often called flat Monte Carlo. Impressively, Flat Monte Carlo achieved the world champion level in Bridge Ginsberg (2001) and Scrabble Sheppard (2002).

Moreover, Monte Carlo rollouts are often used as a part of model predictive control, see Camacho & Alba (2013). As suggested by Chua et al. (2018); Nagabandi et al. (2018), they offer several advantages, including simplicity, ease of parallelization. At the same time, they reach competitive results to other (more complicated) methods on many important tasks. Williams et al. (2016) applied their Model Predictive Path Integral control algorithm (Williams et al., 2015), the approach based on stochastic sampling of trajectories, to the problem of controlling a fifth-scale Auto-Rally vehicle in an aggressive driving task.

Some works aim to compose a planning module into neural network architectures, see e.g., Oh et al. (2017); Farquhar et al. (2017). Kaiser et al. (2019), recent work on model-based Atari, has shown the possibility of sample efficient reinforcement learning with an explicit visual model. Gu et al. (2016) uses model-based methods at the initial phase of training and model-free methods during 'fine-tuning'. Furthermore, there is a body of work that attempts to learn a planning module, see Pascanu et al. (2017); Racanière et al. (2017); Guez et al. (2019).

## 3    METHODS

Reinforcement learning (RL) is formalized with the Markov decision processes (MDP) formalism see Sutton & Barto (2018). An MDP is defined as $(\mathcal{S}, \mathcal{A}, P, r, \gamma)$, where $\mathcal{S}$ is a state space, $\mathcal{A}$ is a set of actions available to an agent, $P$ is the transition kernel, $r$ is reward function and $\gamma \in (0, 1)$ is the discount factor. An agent policy, $\pi : \mathcal{S} \mapsto P(\mathcal{A})$, maps states to distribution over actions. An object central to the MDP formalism is the value function $V^\pi : \mathcal{S} \mapsto \mathbb{R}$ associated with policy $\pi$

$$V^\pi(s) := \mathbb{E}_\pi \left[ \sum_{t=0}^{+\infty} \gamma^t r_t | s_0 = s \right],$$ where $r_t$ denotes the stream of rewards, assuming that the agent operates with policy $\pi$ (which is denoted as $\mathbb{E}_\pi$) and that at $t = 0$ the system starts from $s$. The

objective is to find a policy, which achieves maximal value. In this work we concentrate on planning methods, which in each step search a subspace of the state-space $\mathcal{S}$ to render robust decision.

A Generic Planner, presented in Algorithm 1, gives a unified view on all methods analyzed in the paper: Random Shooting, MCTS and, our novel approach, STS. By a suitable choice of functions SELECT, EXPAND, UPDATE and CHOOSE_ACTION, we can recover each of these methods (see description below).

Typically, a planner is a part of a reinforcement learning (RL) training process, see Algorithm 2. In a positive feedback loop, the planner improves the quality of data used for training of the value function $V_\theta$ and a policy $\pi_\phi$. Conversely, the policy and value function might further improve planning. Implementation details of Algorithm 2 are provided in Appendix A.1.

---

**Algorithm 1** Generic Planner, defines required constants, variables and objects used in further algorithms

| **Require:** | $C$ | planning passes |
|---|---|---|
| | $H$ | planning horizon |
| | $\gamma$ | discount factor |
| **Use:** | $N(s,a)$ | visit count |
| | $W(s,a)$ | total action-value |
| | $Q(s,a)$ | mean action-value |
| | $\mathbf{V}_\theta$ | value function |
| | $\pi_\phi$ | policy |
| | $model$ | environment simulator |

```
# Initialize N, W, Q to zero
function PLANNER(state)
    for 1 . . . C do
        path, leaf ← SELECT(state)
        rollout, leaf ← EXPAND(leaf)
        UPDATE(path, rollout, leaf)
    return CHOOSE_ACTION(state)
```

**Algorithm 2** Training loop, additionally requires environment $env$

```
# Initialize parameters of V_θ, π_φ
# Initialize replay_buffer
repeat
    episode ← COLLECT_EPISODE
    replay_buffer.ADD(episode)
    B ← replay_buffer.BATCH
    Update V_θ, π_φ using B and SGD
until convergence
function COLLECT_EPISODE
    s ← env.RESET
    episode ← []
    repeat
        a ← PLANNER(s)
        s′, r ← env.STEP(a)
        episode.APPEND((s, a, r, s′))
        s ← s′
    until episode is done
    return CALCULATE_TARGET(episode)
```

---

Below we give a detailed description of the planning methods considered in the papers.

**Random Shooting**  In this section we present two instantiations of Algorithm 1, which use Monte Carlo rollouts to evaluate state-actions pairs: Random Shooting and Bandit Shooting, see Algorithm 3 and Algorithm 4, respectively.

---

**Algorithm 3** Random Shooting Planner

```
function SELECT(state)
    s ← state
    a ∼ π_φ(s, ·)
    s′, r ← model.STEP(s, a)
    return (s, a, r), s′

function EXPAND(leaf)
    s_0 ← leaf
    rollout ← (s_k, a_k, r_{k+1})_{k=0}^{H-1}
    where s_{k+1}, r_{k+1} ← model.STEP(s_k, a_k)
    and a_k ∼ π_φ(s_k, ·)
    return rollout, s_H
```

```
function UPDATE(path, rollout, leaf)
    Ĝ ← Σ_{k=1}^{H} γ^{k-1} r_k + γ^H V_θ(leaf)
    where r_k ∈ rollout
    s, a, r ← path
    quality ← r + γ * Ĝ
    W(s, a) ← W(s, a) + quality
    N(s, a) ← N(s, a) + 1
    Q(s, a) ← W(s,a)/N(s,a)

function CHOOSE_ACTION(s)
    return arg max_a Q(s, a)
```

---

**Algorithm 4** Bandit Shooting Planner, additionally requires exploration weight $c_{puct}$

---

**function** SELECT(state)              **function** EXPAND(leaf)
    $s \leftarrow$ state                         Same as in Algorithm 3.
    $U(s,a) \leftarrow \sqrt{\sum_{a'} N(s,a')}/(1 + N(s,a))$    **function** UPDATE(path, rollout)
    $a \leftarrow \arg\max_a(Q(s,a) + c_{puct}\pi_\phi(s,a)U(s,a))$   Same as in Algorithm 3.
    $s',r \leftarrow model.\text{STEP}(s,a)$             **function** CHOOSE_ACTION($s$)
    **return** $(s,a,r),\ s'$                   **return** $\arg\max_a N(s,a)$

---

The simplest version of Algorithm 3, the so-called flat Monte Carlo Ginsberg (2001); Sheppard (2002), does not use a policy $\pi_\phi$ (instead rollouts are uniformly sampled) nor a value function $\mathbf{V}_\theta$ (just truncated sum of rewards $\hat{G} = \sum_{k=1}^{H} \gamma^{k-1} r_k$). The value function is neither used in the experiments with the pre-trained PPO policy in Section 4.2. Bandit Shooting, presented in Algorithm 4, is a Multi-armed Bandits variant of Random Shooting and uses PUCT Silver et al. (2018) rule to improve exploration and thus achieve more reliable evaluations of actions.

---

**Algorithm 5** MCTS, additionally uses tree structure $tree$, requires exploration weight $c_{puct}$ and action sampling temperature $\tau$

---

**function** SELECT(state)               **function** UPDATE(path, rollout, leaf)
    $s \leftarrow$ state                     quality $\leftarrow \mathbf{V}_\theta(\text{leaf})$
    path $\leftarrow []$                     **for** $s,a,r \leftarrow$ reversed(path) **do**
    **while** $s$ belongs to $tree$ **do**           quality $\leftarrow r + \gamma *$ quality
        $a \leftarrow$ SELECT_CHILD($s$)        $W(s,a) \leftarrow W(s,a) +$ quality
        $s',r \leftarrow tree[s][a]$           $N(s,a) \leftarrow N(s,a) + 1$
        path.APPEND($(s,a,r)$)         $Q(s,a) \leftarrow \frac{W(s,a)}{N(s,a)}$
        $s \leftarrow s'$
    **return** path, $s$                 **function** SELECT_CHILD($s$)
                                  $U(s,a) \leftarrow \sqrt{\sum_{a'} N(s,a')}/(1 + N(s,a))$
**function** EXPAND(leaf)               $a \leftarrow \arg\max_a(Q(s,a) + c_{puct}\pi_\phi(s,a)U(s,a))$
    **for** $a \in \mathcal{A}$ **do**               **return** a
        $s',r \leftarrow model.\text{STEP}(\text{leaf},a)$    **function** CHOOSE_ACTION($s$)
        $tree[\text{leaf}][a] \leftarrow (s',r)$        $a \sim \text{softmax}\left(\frac{1}{\tau}\log N(s,\cdot)\right)$
        $W(\text{leaf},a) \leftarrow r + \gamma * \mathbf{V}_\theta(s')$    **return** $a$
        $N(\text{leaf},a) \leftarrow 1$
        $Q(\text{leaf},a) \leftarrow W(\text{leaf},a)$
    **return** $[]$, leaf

---

**Algorithm 6** Shoot Tree Search

---

**function** EXPAND(leaf)            **function** UPDATE(path, rollout, leaf)
    $s \leftarrow$ leaf                   $s' \leftarrow$ leaf
    rollout $\leftarrow []$                $c \leftarrow 1$
    **for** $1 \dots H$ **do**             quality $\leftarrow 0$
        MCTS.EXPAND($s$)        **for** $s,a,r \leftarrow$ reversed(path + rollout) **do**
        $a \leftarrow$ CHOOSE_ACTION($s$)     **if** $s' \in$ path **then**
        $s',r \leftarrow tree[s][a]$          v $\leftarrow 0$
        rollout.APPEND($(s,a,r)$)     **else**
        $s \leftarrow s'$                    v $\leftarrow \mathbf{V}_\theta(s')$
    **return** rollout, $s$           $c \leftarrow c + 1$
                            quality $\leftarrow c * r + \gamma * (\text{quality} + \text{v})$
**function** SELECT(state)         $W(s,a) \leftarrow W(s,a) +$ quality
    Same as in Algorithm 5.      $N(s,a) \leftarrow N(s,a) + c$
**function** CHOOSE_ACTION($s$)     $Q(s,a) \leftarrow \frac{W(s,a)}{N(s,a)}$
    Same as in Algorithm 5.      $s' \leftarrow s$

---

**MCTS** Monte Carlo Tree Search (MCTS) is a family of methods, that iteratively and explicitly build a search tree, see Browne et al. (2012). It follows the schema of Algorithm 1. SELECT traverses down the tree, according to an in-tree policy, until a leaf is encountered. EXPAND grows the tree by

adding the leaf's children. The values of these new nodes are estimated, usually with the help of a rollout policy in a similar vein as Random Shooting Planner, or via the value network $V_\theta$ (see Silver et al. (2017)). In this work, we refer to the latter version, using value networks, as MCTS. Finally, `UPDATE` backpropagates these values from the leaf up the tree. After planning, `CHOOSE_ACTION` chooses an action to take by sampling from the empirical child visitation distribution, sharpened by a predefined temperature parameter $\tau$. This is consistent with MuZero, as used in the Atari domain (Schrittwieser, 2019). A basic variant of MCTS is presented in Algorithm 5. More details are provided in Appendix A.5.

**Shoot Tree Search**   Shoot Tree Search (STS) extends MCTS in a novel way, by redesigning the expansion phase, see Algorithm 6. Given a leaf and a planning horizon $H$ the method expands $H$ consecutive vertices starting from the leaf. Each new node is chosen according to the in-tree policy and is added to the search tree. Note a crucial difference between STS and vanilla MCTS using random rollouts: in contrast to the latter, STS adds visited nodes to the tree, so the explored paths can easily be branched out during later planning passes. We call this mechanism *multi-step expansion*. Intuitively, *multi-step expansion* tilts slightly the search towards DFS. Its advantage comes from making "faster advances" towards the solution, though possibly at risk of missing some promising nodes. Our experiments support these intuitions, suggesting a sweet-spot around $H = 10$ (see experiments in Table 4.1 and Section A.9.1).

A similar method was proposed in Soemers et al. (2016), with two crucial differences. First, Soemers et al. (2016) uses hard-coded heuristics while we embed STS into the RL training. Second, in Soemers et al. (2016) only the last estimate of value is backpropagated. Our `UPDATE` backpropagates an aggregate value estimates calculated on the rollout of *multi-step expansion*. We weight this update by the rollout length (hence $N(s, a) \leftarrow N(s, a) + c$). We consider our method more natural and, importantly, it presents better experimental results; see also Section A.9.4.

STS can be viewed as a sophisticated version of Random Shooting applied to MCTS. In this interpretation, STS interpolates between the two methods. We demonstrate empirically that the change introduced by STS is essential to solving challenging RL domains; see Section 4. We note that $H = 1$ corresponds to MCTS.

Interestingly, in some of our experiments, we identified that the tree traversal performed during `SELECT` was the computational bottleneck. The cost of building the search tree is quadratic with respect to its depth. STS allows to significantly reduce this cost since a single tree traversal adds not one but $H$ new nodes. To get this computational benefit we tweak `UPDATE` to backpropagate all values from `leaf` and `rollout` in one pass. A more formal analysis of computational gains is presented in Lemma A.6.1.

Equipping STS with additional exploration mechanism (see Appendix A.5, A.4, and A.6), can guarantee that every state action pair will be visited infinitely often. Combining this with Robbins-Monro conditions for learning rate, implies the convergence in tabular case, following the classical tabular Q-learning convergence theorem (see Tsitsiklis (1994)).

## 4 EXPERIMENTS

We tested the spectrum of algorithms presented in Section 3 on the Sokoban and Google Research Football domains. Those tasks present numerous challenges, which evaluate various properties of planning algorithms. Our experiments support the hypothesis that STS build a more efficient search tree. This is strengthen by measuring the tree size in isolation (see Table 1), its average depth (see Section A.9.1) and comparisons to AlphaGo (see Section A.9.2). For the training details, a list of hyper-parameters and network architectures are presented in Appendix A.1, Appendix A.2 and Appendix A.3 respectively.

We stress that in all comparisons, we set parameters $C, H$ so that MCTS and STS perform the same number of node expansions during each `PLANNER` call; see Algorithm 1. This ensures the same computational power requirements (e.g., number of neural network evaluations) and similar memory usage.

In Appendix A.10, we present two thought experiments, supported by formal proofs, where we argue that STS can better handle certain errors in value functions by using the *multi-step expansion*. We

show that STS can get quicker to the regions with the accurate values during planning and it is less prone to entering 'decoy' paths. The errors are inevitable during training and when using function approximators.

## 4.1 SOKOBAN

Sokoban is a well-known combinatorial puzzle, where the agent's goal is to push all boxes (marked as yellow, crossed squares) to the designed spots (marked as squares with a red dot in the middle), see Figure 1. Additionally, to the navigational challenge, Sokoban's difficulty is attributed to the irreversibility of certain actions. A typical example is pushing a box into a corner, though there are multiple less apparent cases. The environment's complexity is formalized by the fact that, deciding whether a level of Sokoban is solvable or not, is PSPACE-complete, see e.g. Dor & Zwick (1999). Due to these challenges, the game is often used to test reinforcement learning and planning methods.

| Scenario | C | H | S. rate | $N_p$ | $N_t$ | $N_g$ |
|---|---|---|---|---|---|---|
| av. loops | 256 | 1 | 95.2% | 1224 | 1224 | 716 |
| | 64 | 4 | 96.5% | 299 | 1194 | 830 |
| | 16 | 16 | 95.7% | 114 | 1822 | 1333 |
| | 4 | 64 | 89% | 62 | 3960 | 1491 |
| no av. loops | 256 | 1 | 84.5% | 1497 | 1497 | 376 |
| | 32 | 8 | 88.4% | 185 | 1483 | 409 |
| | 2 | 128 | 65.3% | 36 | 4589 | 967 |

Table 1: Comparison on evaluation of MCTS and STS. $C, H$ are parameters in Algorithm 1. S. rate is the ratio of solved boards, $N_p, N_t(= N_p \cdot H), N_g$ are the average number of passes, tree nodes and game states observed until the solution is found. Full table is available in Table 4.

We use procedurally generated Sokoban levels, as proposed by Racanière et al. (2017). The agent is rewarded with 1 by putting a box into a designated spot and additonally with 10 when all the boxes are in place. We use Sokoban with the board of size $(10, 10)$, 4 boxes, and the limit of 200 steps. We use an MCTS implementation with transposition tables and a loop avoidance mechanism, see Miłoś et al. (2019) and Appendix A.5.

In the first experiment, we evaluated the planning capabilities of STS in isolation from training. To this end, we used a pre-trained value function (we used MCTS to get this) and varied the number of passes $C$ and the depth of multi-step expansion $H$, such that $H \cdot C$ remains constant. In this way, we ensure a fair comparison because the same computing power is used. In Table 1 we present quantities $(N_p, N_t, N_g)$, which measure planning costs. Arguably, the most important of these is $N_t$, which denotes the total size of the planning tree used to find the solution. In the two presented scenarios, there is a sweet spot for the choice of $H$. For this choice, the number of tree nodes, $N_t$, is the smallest, and even more importantly, we observe an increase in the solved rate. This may possibly be explained by the fact that the number of distinct visited game states, $N_g$, grows. This suggests that STS explores more aggressively and efficiently.

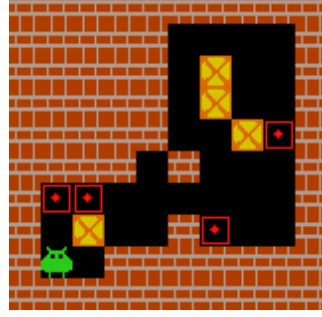

Figure 1: Example $(10, 10)$ Sokoban board with 4 boxes. Boxes (yellow) are to be pushed by agent (green) to designed spots (red). The optimal solution in this level has 37 steps.

In the second line of experiments, we analyzed the training performance (see Algorithm 2). For MCTS we used $C = 50$ passes per step, while for STS we considered $C = 10$ passes with multi-step expansion $H = 5$. The learning curve for STS dominates the learning curve for MCTS, which persists throughout training, see Figure 2. Since the difficulty of Sokoban levels increases progressively, the achieved improvement is substantial, even though in absolute terms, it may seem small.

To better understand where the differences in performance stem from, we evaluated MCTS with 50 passes and STS with 10 passes and 5 steps of multi-step expansion, both without the avoid loops mechanism. We found that 68% of boards is solved by both MCTS and STS, 10% only by STS, and 2.1% only by MCTS. On several examples, we observed that STS could recover better than MCTS from errors in value function, which are relatively localized in the state space, even though they might be quite significant in value. This can be attributed to multi-step expansion, which exits from the erroneous region more quickly by correcting biased value functions estimates with deep search (note that deep paths, while introducing more variance, will stronger discount biased value function).

The downside is that sometimes STS is overoptimistic, pushing into dead-end states. We include the detailed analysis on the example room in the Appendix A.7.3.

Methods based on random shooting perform poorly for Sokoban: we evaluated Bandit Shooting (Algorithm 4), which struggled to exceed 5% solved rate. Only when the difficulty of boards was significantly reduced, to the board size of $(6, 6)$ with 2 boxes, this method achieved results above 90%. Our shooting setup included applying loop avoidance improvements. This feature is highly effective in the case of MCTS (and STS) but did not bring much improvement for shooting methods. Details are provided in Appendix A.7.2.

We conclude with a conjecture that for domains with combinatorial complexity, tree methods (MCTS or STS) significantly outperform shooting methods, and STS offers some benefits over MCTS.

STS is predominantly meant to improve search for small computational budgets. In Appendix A.10.1 we present also result for a ten times bigger budget of $H \cdot C = 500$.

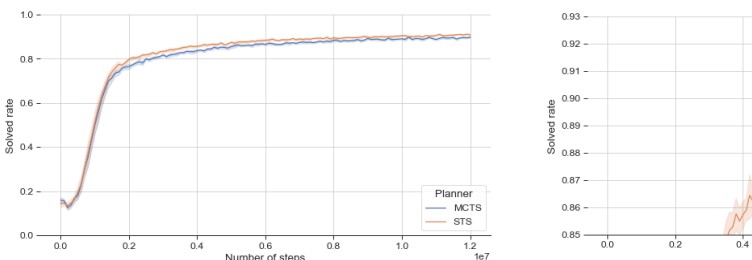

Figure 2: Learning curve for Sokoban domain. Left figure shows full results, right one inspects the same data for limited interval of values on the y axis. The results are averaged over 10 runs, shaded areas shows 95% confidence intervals. The x axis is the number of collected samples.

## 4.2 GOOGLE RESEARCH FOOTBALL

Google Research Football (GRF) is an environment recently introduced in Kurach et al. (2019). It is an advanced, physics-based simulator of the game of football. It is designed to offer a set of challenges for testing RL algorithms. At the same time, it is highly-optimized and open-sourced. GRF is modeled after popular football (a.k.a. soccer) video games, fun and engaging for humans. As such, it requires both tactical and strategical decision-making. This makes it an interesting benchmark for planning algorithms. A part of GRF is the Football Academy consisting of 11 scenarios highlighting various tactical difficulties, see Kurach et al. (2019, Table 10) for description. Due to its diversity, the GRF Academy is an excellent testing ground of planning methods listed in Section 3, including STS. GRF provides several state representations, including internal game representation as well as visual observation. We tested both of them: the former was processed with an MLP architecture, while the latter with a convolutional neural network. Details are provided in Appendix A.1.

One feature which makes GRF hard (and thus interesting) for planning is its relatively large action space (19 actions). From the perspective of the design of a low budget planner, this can be viewed as a challenge.

A GRF Academy episode is considered finished after 100 steps or when the goal is scored by the agent. The game is stochastic, hence we report the solved rates over at least 20 episodes per environment. In Table 2 we compare STS with various other methods. To the best of our knowledge, no prior work has evaluated model-based methods on Google Research Football. Hence we provide two baselines: model-free PPO results, reported by the authors of GRF, and model-based AlphaZero implemented by us, with a minor environment-specific modification. To efficiently deal with the large action space, we use a Q-network $Q(s, a)$ instead of a value network $V(s)$ to evaluate all actions at the same time when expanding a leaf. We provide an ablation on this architectural choice in Appendix A.9.

We report the median of the solved rates in at least three runs with different seeds.

**Random shooting** For each of the Random Shooting and Bandit Shooting planners (Algorithm 3 and Algorithm 4, respectively), we performed two batches of experiments: with and without training.

The former used two different state representation and, as a consequence, two different architectures (MLP and Conv.). The latter used a uniform policy (flat) or a pre-trained policy (PPO). For all the variants, we set $C = 30$ passes and a planning horizon $H = 10$. More details can be found in Appendix A.1.

The flat version cannot solve GRF Academy tasks. This is rather unsurprising and confirms that it is a challenging test suite. The Bandit Shooting algorithm generally offers a better performance both when using the pretrained policy or training from scratch. This indicates that bandit-based exploration results in more reliable estimates of action values. Bandit Shooting Conv. experiments are better than the baseline in 6 cases and worse in 4. This shows that, at least in some environments, planning can improve performance. We also tested whether mixing the policy with Dirichlet noise

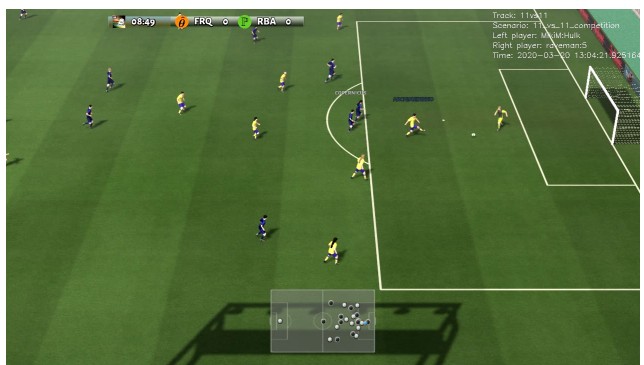

Figure 3: Example from the Google Football League

(see Silver et al. (2018)) and sampling an action to take in an environment can impact exploration and training performance. Nevertheless, the results were inconclusive (see Appendix A.4 for details). It can be seen that the *corner* scenario is particularly challenging: the baseline scores on the lower end of the spectrum, the shooting algorithms rather underperformed and the training was quite unstable. The results improved significantly under the STS algorithm. In the Shooting experiments, we used approx. 1.5M training samples (median).

**STS and MCTS**   STS achieves state-of-the-art results on the GRF Academy and significantly outperforms other methods. For STS we used $C = 30$ passes with $H = 10$ and for MCTS we set corresponding $C = 300$.

STS Conv. completely solves 8 out of 11 academy environments and is the best or close to the best on the remaining 3. One can observe that in Corner, Counterattack easy and hard, Pass and shoot with keeper, Run to score with keeper, and Single goal vs. lazy academies the difference between STS and MCTS, as well as most of the other methods, is substantial. See Table 2 for exact results. We stress that STS Conv. easily beats any other method in one-to-one comparison across all academies. STS MLP achieves a close second place. These results provide further evidence that STS gives a boost in environments requiring long-horizon planning. This stands in sharp contrast with MCTS, which was not able to achieve impressive results in the considered time budget. We found that exploration was a challenge in GRF Academy environments. Namely, training often got stuck in disadvantageous regions of the state space, which was caused by unfavorable random initialization of the value function. To deal with it, the last layer of the value function neural network was initialized to 0. We suspect this zero-initialization method might be useful in other domains as well. Our findings is consistent with recent recommendations of (Andrychowicz et al., 2020, Section 3.2) given in the model-free setting. In the STS experiments we used approx. 0.8M training samples (median).

More details can be found in Appendix A.8, including ablations. They indicate that *multi-step expansion* of STS blends well with various elements of the MCTS toolbox as well as demonstrate the impact of the aforementioned zero-initialization.

## 5   CONCLUSIONS AND FURTHER WORK

In this paper, we introduced a new algorithm, Shoot Tree Search. STS aims to explicitly address the dilemma between depth and breadth search in large state spaces. That touches upon interesting issues of using randomness and structure in search algorithms. The core improvement is *multi-step expansion*, which may be used to control the depth of search and inject into planning more randomness via random multi-step expansions. Having empirically verified the efficiency of this extension in many challenging scenarios, we argue that it could be included in a standard MCTS toolbox. We

| Method | | 3 vs. 1 with keeper | Corner | Counterattack easy | Counterattack hard | Empty goal | Empty goal close | Pass and shoot with keeper | Run pass and shoot with keeper | Run to score | Run to score with keeper | Single goal versus lazy |
|---|---|---|---|---|---|---|---|---|---|---|---|---|
| | PPO | 0.90 | 0.10 | 0.70 | 0.65 | 0.90 | **1.00** | 0.65 | 0.90 | 0.90 | **1.00** | 0.90 |
| Random Shooting | flat | 0.10 | 0.00 | 0.05 | 0.10 | 0.00 | 0.95 | 0.05 | 0.10 | 0.00 | 0.00 | 0.00 |
| | PPO | 0.45 | 0.10 | 0.10 | 0.30 | **1.00** | **1.00** | 0.25 | 0.80 | 0.80 | 0.20 | 0.30 |
| | MLP | 0.90 | **0.87** | 0.80 | 0.73 | 0.93 | **1.00** | 0.87 | 0.70 | 0.90 | 0.37 | 0.67 |
| Bandit Shooting | flat | 0.20 | 0.10 | 0.00 | 0.00 | 0.05 | 0.35 | 0.05 | 0.05 | 0.05 | 0.00 | 0.00 |
| | PPO | **1.00** | 0.05 | 0.95 | 0.80 | **1.00** | **1.00** | 0.55 | **1.00** | 0.85 | 0.45 | 0.60 |
| | MLP | 0.87 | 0.47 | 0.73 | 0.60 | **1.00** | **1.00** | 0.90 | 0.80 | 0.93 | **1.00** | 0.60 |
| | Conv. | 0.97 | 0.41 | 0.81 | 0.44 | 0.97 | **1.00** | 0.94 | 0.69 | **1.00** | 0.91 | 0.00 |
| MCTS | Conv. | 0.81 | 0.50 | 0.31 | 0.31 | 0.99 | **1.00** | 0.45 | 0.89 | 0.70 | 0.00 | 0.00 |
| STS | MLP | **1.00** | 0.78 | **1.00** | 0.97 | **1.00** | **1.00** | 0.94 | 0.97 | **1.00** | 0.94 | 0.94 |
| | Conv. | **1.00** | 0.81 | **1.00** | **1.00** | **1.00** | **1.00** | **1.00** | **1.00** | **1.00** | 0.97 | **0.97** |

Table 2: Summary of selected algorithms' performance on GRF. Entries are rounded solved rates. PPO results come from Kurach et al. (2019).

speculate that the effectiveness of STS stems from a better balance between breadth and depth search. While our experiments support this claim, we plan a more detailed analysis, possibly using methods developed in James et al. (2017).

There are many interesting follow-up research directions involving STS. One of them concerns the automatic choice of the multi-step expansion depth, $H$, during training. This could not only improve the performance of the method, but also alleviate the necessity for fine-tuning this additional hyper-parameter. Another, quite natural extension of this work is to use learned models. As a research question, this typically splits into two sub-problems: learn an accurate model, or adjust the planner to accommodate for the model's deficiencies. An exciting research avenue, is related to a multi-agent version of GRF. This constitutes an open challenge both for planning and learning models.

It is interesting to study STS itself. Historically, the fusion of different multi-step estimates (such as TD($\lambda$) or GAE) lead to significant improvements, and it is only natural to ask if a similar advancement can be reached here. Moreover, STS could be combined with different statistical tree search methods, where statistics other than the expected value (e.g. max) are stored and updated (see e.g. Agostinelli et al. (2019)). The method could also be augmented with uncertainty estimation (e.g. in the spirit of Miłoś et al. (2019)) to strengthen the exploration, and consequently the algorithm.

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

## A.1 TRAINING DETAILS

We provide the code of our methods and hyper-parameters configuration files in `https://github.com/shoot-tree-search/sts`.

The training loop follows the logic of Algorithm 2. We use a distributed setup with 30 workers and a replay buffer of size 30000. We perform 1000 optimizer updates on batches of transitions whenever all workers collect and store one full episode. During batch sampling, we ensured an equal amount of examples from solved and unsolved episodes. In GRF and Sokoban experiments, each episode was limited to 100 and 200 time steps, respectively.

A value function approximator, $\mathbf{V}_\theta$, is trained via the MSE loss using targets calculated by `CALCULATE_TARGET`. In shooting experiments we use "reward-to-go" targets $\sum_{i=t+1}^{T} \gamma^{i-t-1} r_i$, where $T$ is the terminal time-step in an episode. In MCTS and STS in GRF experiments (see Section A.5 for details) we use "tree action-values" targets, similar to the one used in Hamrick et al. (2020); Miłoś et al. (2019).

Policy, $\pi_\phi$, is trained using the cross-entropy loss. As targets, we use one-hot encoded actions chosen in the environment for Random Shooting and the empirical distribution of actions chosen in the root during the planning for Bandit Shooting, MCTS, and STS.

The total loss is a weighted sum of the value function (or the $Q$-function) loss, the policy loss (weighted by $1e{-}2$ in Random Shooting and Bandit Shooting, and $1e{-}3$ in MCTS and STS), and a regularizing, $l_2$ term (weighted by $1e{-}6$).

A pre-trained PPO policy in Shooting methods was obtained using a script included in the Google Research Football repository (see Kurach et al. (2019)) and the OpenAI Baselines (Dhariwal et al. (2017)) PPO2 implementation.

## A.2 HYPER-PARAMETERS

Table 3 presents hyper-parameters used in our experiments. These were based on hyper-parameters previously proposed in the literature and substantial amount of tuning experiments ($> 3000$).

## A.3 NETWORK ARCHITECTURES

In GRF experiments we use two different state representations: 'simple115' and 'extended' (see Section A.8). In the former case, we use an MLP architecture with two hidden layers of 64 neurons, while in the latter case, we use 4 convolutional layers with 16, 3x3, filters, zero-padding and stride 2, followed by a dense layer of 64 neurons. In both cases, two heads, corresponding to a value function (or $Q$-function for MCTS and STS) and policy, follow.

In Sokoban experiments, we use 5 convolutional layers of 64, 3x3, filters with zero-padding and stride 1, followed by a dense layer of 128 neurons and heads corresponding to a value function and policy (policy is used only for Shooting methods).

In all the cases, we use the ReLU non-linearity. We use the standard Keras initialization schemes, except for MCTS and STS in GRF experiments, see Section A.8.2.

| Parameter | Sokoban | | | Google Research Football | | |
|---|---|---|---|---|---|---|
| | Shooting | MCTS | STS | Shooting | MCTS | STS |
| Number of passes $C$ | 48 | 50 | 10 | 30 | 300 | 30 |
| Planning horizon $H$ | 5 | 1 | 5 | 10 | 1 | 10 |
| Discounting $\gamma$ | 0.99 | 0.99 | 0.99 | 0.95 / 0.99[1] | 0.99 | 0.99 |
| Exploration weight $c_{puct}$ | 10.0[2] | 0.0 | 0.0 | 1.0 / 2.5[3] | 1.0 | 1.0 |
| Policy $\pi_\phi$ temperature[4] | 2.0 | - | - | 2.0 | 1.0 | 1.0 |
| Action sampling temp. $\tau$ | - | - | - | 0.3[5] | 0.3 | 0.3 |
| Dirichlet parameter $\alpha$ | - | - | - | 0.03[5] | 0.3 | 0.3 |
| Noise weight $c_{noise}$ | - | - | - | 0.1[5] | 0.1 | 0.1 / 0.3[6] |
| Depth limit `depth limit`[7] | - | - | - | - | 30 | 30 |
| VF zero-initialization[8] | no | no | no | no | yes | yes |
| Optimizer | RMS | RMS | RMS | RMS | Adam | Adam |
| Learning rate | 2.5e−4 | 2.5e−4 | 2.5e−4 | 1.0e−4 | 1.0e−3 | 1.0e−3 |
| Batch size | 32 | 32 | 32 | 64 | 64 | 64 |
| Target function[9] | *rew2goT* | *treeT* | *treeT* | *rew2goT* | *treeT* | *treeT* |

[1] All $\gamma = 0.99$ except for Shooting experiments with a uniform and a pre-trained PPO policy, where $\gamma = 0.95$.
[2] Applies only to Bandit Shooting.
[3] $c_{puct} = 1.0$ for Bandit Shooting with a uniform and a pre-trained PPO policy and $c_{puct} = 2.5$ for Bandit Shooting with a trained policy.
[4] Softmax temperature. MCTS and STS in Sokoban does not use policy, see Section A.5 for details.
[5] Applies only to Bandit Shooting with additional exploration mechanisms, see Section A.4.
[6] $c_{noise} = 1.0$ for STS Conv. and $c_{noise} = 0.3$ for STS MLP.
[7] The maximum number of nodes visited in a single planning pass, see Section A.5.
[8] If the last layer of a value function neural network was initialized to 0, see Section A.8.2.
[9] Indicates how training targets (CALCULATE_TARGET in Algorithm 2) are obtained. *rew2goT* and *treeT* corresponds "reward-to-go" and "tree action-values" described in Section A.1.

Table 3: Default values of hyper-parameters used in our experiments.

## A.4 BANDIT SHOOTING

**Algorithm 7** Bandit Shooting Planner with additional exploration mechanisms, requires exploration weight $c_{puct}$, action sampling temperature $\tau$, noise weight $c_{noise}$ and Dirichlet distribution parameter $\alpha$

```
function SELECT(state)                          function EXPAND(leaf)
    s ← state                                       The same as in Algorithm 3.
    P(s, a) ← (1 − c_noise)π_φ(s, a) + c_noise D   function UPDATE(path, rollout)
    U(s, a) ← √(Σ_a' N(s, a'))/(1 + N(s, a))        The same as in Algorithm 3.
    a ← arg max_a (Q(s, a) + c_puct P(s, a)U(s, a)) function CHOOSE_ACTION(s)
    s', r ← model.STEP(s, a)                        a ∼ softmax(1/τ log N(s, ·))
    return (s, a, r), s'                            return a
```

Algorithm 7 describes Bandit Shooting with additional exploration mechanisms: mixing the policy with Dirichlet noise (as in Silver et al. (2018)) and action sampling with temperature $\tau$ in CHOOSE_ACTION($s$). The noise variable $D$ is sampled from the Dirchlet distribution $Dir(\alpha)$ each time when PLANNER is called (see also Algorithm 2).

## A.5 MCTS

In our experiments, we used various implementations of MCTS. The reasons were two-fold. First, some implementation details fit better Sokoban and some GRF. Second, we wanted to check in various cases that the multi-step expansion is beneficial, see Section A.6.

In Sokoban experiments, we used the MCTS implementation similar to the one in Miłoś et al. (2019), containing a loop avoidance mechanism and transposition tables. The loop avoidance mechanism

alters `SELECT` and `CHOOSE_ACTION` (see Algorithm 5) so that the selected `path` does not contain repetitions of states. The transposition tables are a rather standard technique, which proposes to accumulate search statistics (i.e., $W, N, Q$) for states of the environment (rather than for the nodes of the search tree, as it happens in the standard case).

In GRF, we used our custom implementation of MCTS based on the one in Silver et al. (2017). It uses leaf evaluation with $Q$-function and policy networks. The $Q$-function is used to evaluate all children of a given node at once (instead of separately invoking value function $\mathbf{V}_\theta$ in `UPDATE`). The policy network is considered to be 'prior' for choosing actions, similarly as in `SELECT` in Algorithm 7. Dirichlet noise, parameterized by $\alpha$ and $c_{noise}$, is mixed with the prior in the root and action sampling with temperature $\tau$ is used to choose action on the real environment, similarly as in Bandit Shooting with additional exploration mechanisms in Section A.4. Additionally, we put a limit, `depth limit`, on the maximum number of nodes visited in a single STS pass.

## A.6   STS

We tested STS with two MCTS setups described in Section A.5. In both the cases we observed substantial experimental improvements as reported in Section A.7 and Section A.8. This alone, in our view, provides enough evidence that the *multi-step expansion* is a useful method.

Apart from this, STS offers practical computational benefits, which are analyzed below.

### A.6.1   COMPUTATIONAL BENEFITS OF STS

We distinguish three types of computational costs in MCTS (see Algorithm 5):

1. Traversing down the search tree (performed in `SELECT` and `EXPAND`).
2. Backpropagation of values and counts update (handled by `UPDATE`).
3. Evaluation of heuristics (value network $\mathbf{V}_\theta$, or $Q$-function and policy as described in Section A.5)

In large GRF experiments, we found that it was the first cost that dominated the remaining two. The reason is that the cost of building a search tree is quadratic to its depth. The use of *multi-step expansion* significantly reduces this cost as several nodes are added during single tree traversal. In our case, these benefits allowed for much smoother experimenting with GRF and are, arguably, a step towards developing more efficient planners. We expect this might be practically useful (i.e., costs 1 and 2 are dominant) when the search size is large, or the heuristic evaluation is relatively cheap compared to the environment step. This is the case in some of our GRF experiments. The GRF simulator is rather complex and slower than small MLP networks.

The following simple lemma offers some theoretical analysis.

**Lemma A.6.1.** *Assume that STS and MCTS build the same tree $\mathcal{T}$, starting from the root state $s_0$. Denote the number of nodes in $\mathcal{T}$ as $C$ and the number of nodes to be added at a single multi-step expansion of STS as $H$. Then the number of steps in $\mathcal{T}$ performed by STS will be lower compared to MCTS by a factor in $[\frac{h-1}{2}, h]$.*

*Proof.* Lets consider $h$ consecutive nodes $s_1, \ldots, s_h$ in the search tree added in a single `EXPAND` step during STS search. In STS, the number of steps, $C_{STS}$, in the tree during `SELECT` and `EXPAND` is equal to $h+d$, where $d$ is distance between $s_0$ and $s_1$ in $\mathcal{T}$. To add the same set of nodes during MCTS search, one need $h$ separate calls to `SELECT` and `EXPAND`. The total number of steps performed is $C_{MCTS} = \sum_{k=0}^{h-1} d + k + 1 = hd + h\frac{h-1}{2}$. Clearly,

$$\frac{h-1}{2} C_{STS} \leq C_{MCTS} \leq h C_{STS}.$$

Similar calculation hold for the costs of backpropagation. □

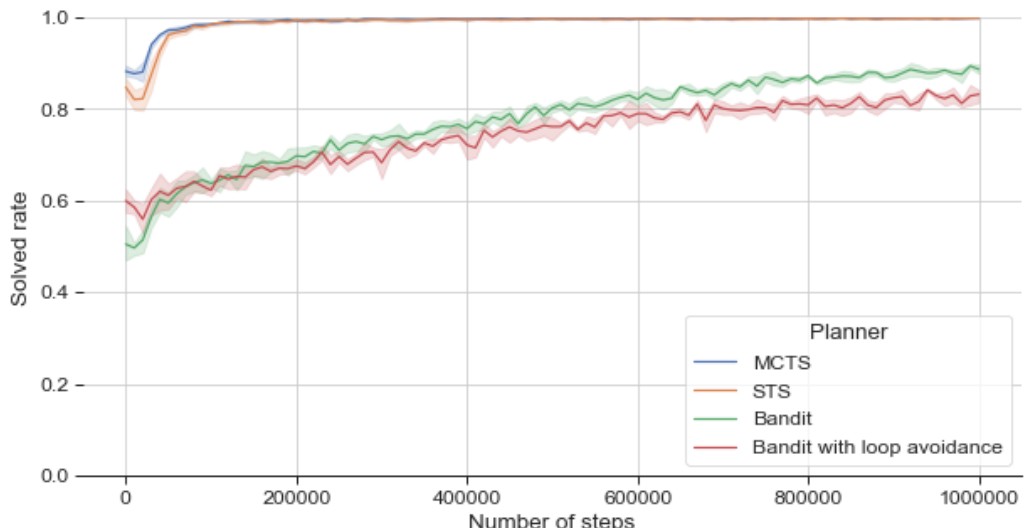

Figure 4: Sokoban on simpler boards: training curves for MCTS, STS and Bandit Shooting with and without loop avoidance. Mean over 5 seeds with shaded regions representing $95\%$ confidence intervals.

## A.7 SOKOBAN EXPERIMENTS

For a description of Sokoban see Section 4.1. In our experiments, we used inputs of dimension $(x, x, 7)$, where $(x, x)$ is the size of the board ($(10, 10)$ in most cases) and $7$ is one-hot encoding of the state of a given cell (enumerated as follows: wall, empty, target, box_target, box, player, player_target). In most experiments, we used $4$ boxes. The agent is rewarded with $1$ by putting a box into a designated spot and additionally with $10$ when all the boxes are in place[1]. The action space consists of four movement directions (up, down, right, left).

### A.7.1 EVALUATION EXPERIMENTS

In Table 4 we show full details of the evaluation experiment (which complements Table 1). Recall that in this experiment, we evaluated the planning capabilities of STS in isolation from training. To this end, we used a pre-trained value function and varied the number of passes $C$ and the depth of multi-step expansion $H$, such that $H \cdot C$ remains constant. In Table 4, we present quantities $(N_p, N_t, N_g)$, which measure planning costs for finding a solution (the average number of passes, tree nodes and game states observed, respectively, until the solution is found). We run experiments with and without the loop avoidance mechanism (see Section A.5). We observe that there is a sweet spot for the choice of $H$. It is evident for the 'no avoid loop' case, $C = 32, H = 8$. For this choice, the number of tree nodes, $N_t$, which is the most important metric, is the smallest. Interestingly, we observe a significant increase in the solved rate. This may be explained by the fact that the number of distinct visited game states, $N_g$, grows. This suggests that STS explores more aggressively and efficiently. For bigger $H$, we observe a further increase of the solved rate until some point, though at the cost of much bigger $N_t$.

In experiments with the avoid loop mechanism, there is a similar effect for $C = 64, H = 4$, though more subtle, probably because results are already quite strong. Moreover, we observe a more significant drop in performance as $H$ increases (when planning resembles more shooting methods).

The values presented in Table 4 are averages over more than $5000$ boards.

---

[1]Our Sokoban code is fully compatible with Racanière et al. (2017).

| Scenario | C | H | S. rate | $N_p$ | $N_t$ | $N_g$ |
|---|---|---|---|---|---|---|
| avoid loops | 256 | 1 | 95.2% | 1224 | 1224 | 716 |
| | 128 | 2 | 95.9% | 569 | 1137 | 728 |
| | 64 | 4 | 96.5% | 299 | 1194 | 830 |
| | 32 | 8 | 95.9% | 173 | 1385 | 1040 |
| | 16 | 16 | 95.7% | 114 | 1822 | 1333 |
| | 8 | 32 | 93.4% | 79 | 2527 | 1528 |
| | 4 | 64 | 89% | 62 | 3960 | 1491 |
| | 2 | 128 | 80% | 52.7 | 6754 | 1207 |
| no avoid loops | 256 | 1 | 84.5% | 1497 | 1497 | 376 |
| | 128 | 2 | 86.3% | 724 | 1448 | 332 |
| | 64 | 4 | 87.8% | 385 | 1541 | 370 |
| | 32 | 8 | 88.4% | 185 | 1483 | 409 |
| | 16 | 16 | 89.5% | 110 | 1754 | 539 |
| | 8 | 32 | 89.9% | 84 | 2690 | 882 |
| | 4 | 64 | 85.2% | 68 | 4463 | 1300 |
| | 2 | 128 | 65.3% | 36 | 4589 | 967 |

Table 4: Evaluation of various STS settings on Sokoban

### A.7.2 MCTS AND SHOOTING ON SIMPLER BOARDS

We found the Bandit Shooting method underperforming on Sokoban. As a sanity test, we tested a simpler setting with smaller boards of size $(6, 6)$ and two boxes. Learning curves are presented in Figure 4. MCTS and STS experiments quickly learn to solve over $99\%$ of boards. Bandit Shooting experiment showed stable but much slower progress. We also evaluated the version of Bandit Shooting, with additional loop avoidance, see Section A.5. This mechanism was beneficial for MCTS and STS but failed to bring improvements for the shooting algorithms.

### A.7.3 CORRECTING BIASED VALUE FUNCTIONS ESTIMATES WITH DEEP SEARCH

To generate value function heatmaps we evaluated the pre-trained MCTS value function for each possible agent position in a room. Figures 5 and 6 present two chosen rooms with their corresponding VF heatmaps. Specifically, the room in Figure 6 was solved by the STS with 10 passes and 5 steps of multi-step expansion and wasn't solved by the MCTS with 50 passes, both without the avoid loops mechanism. We include movies of both agents in this room in the code repository: `https://github.com/shoot-tree-search/sts/tree/master/movies`.

Because the value function is biased, it makes MCTS stuck in states with overestimated value. See Figure 6, in this room MCTS gets stuck in the bottom-left region. However, with a deeper tree, STS can get unstuck quicker and still find a solution. Remember that the search statistics (i.e., $W, N, Q$) are accumulated for states of the environment (see Appendix A.5). As this overestimated region gets searched deeper the bias in the value function gets discounted more and the agent figures out there are no rewards in reality. At some point, other actions will have a higher value and the agent has a chance to get unstuck and explore other parts of the room. That being said, it should be noted that this "potential well" will still attract the agent, make its planning paths distracted, even when it gets unstuck. STS is less vulnerable to this effect and is able to solve this room despite high bias in the value function.

## A.8 GOOGLE RESEARCH FOOTBALL EXPERIMENTS

For a description of Google Research Football see Section 4.2. A Google Research Football academy environment is considered solved when an agent scores a goal. Reported results correspond to solved rates over 20 episodes in case of Shooting methods with an uniform and a pre-trained policy and around 30 episodes in case of all other methods. Results for MCTS, STS, and Shooting methods

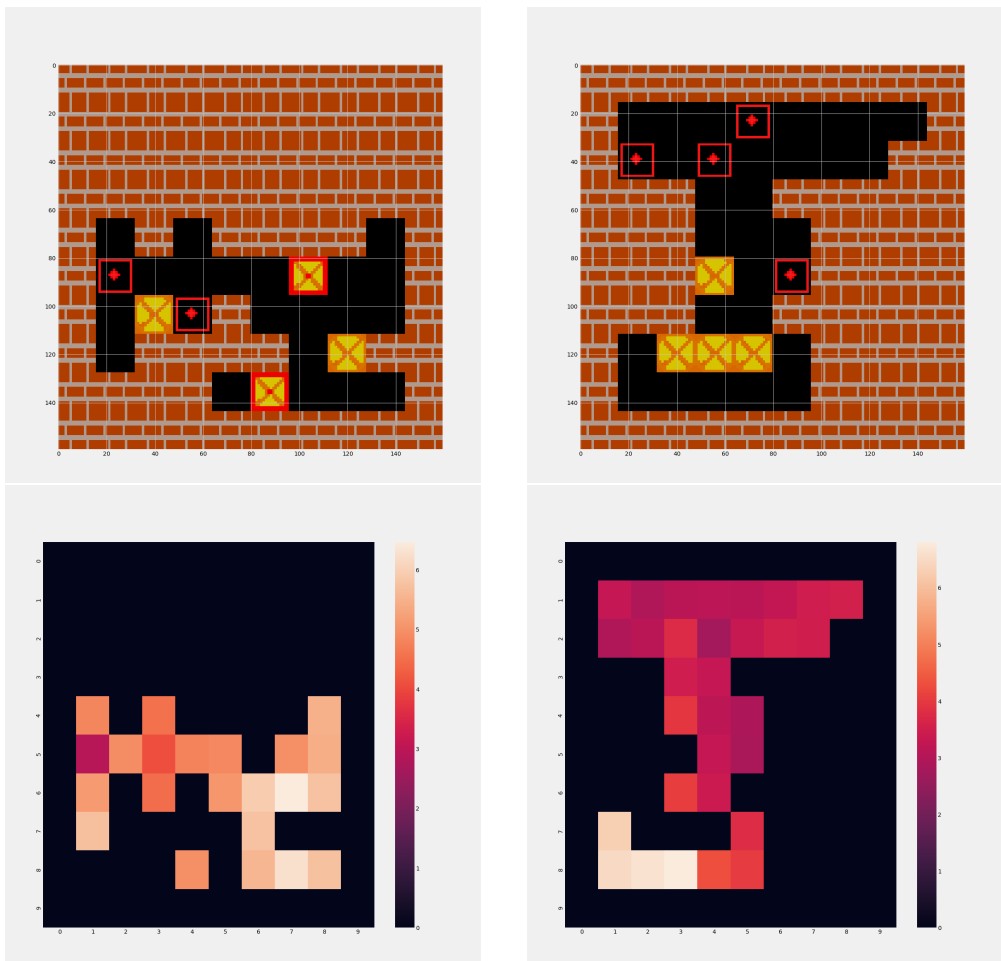

Figure 5: Sokoban value function heatmap, brighter means higher value estimate.

Figure 6: Sokoban value function heatmap, brighter means higher value estimate.

with the trained policy are medians of at least three training runs. During evaluations we disabled Dirichlet noise and action sampling (in Bandit Shooting Expl., MCTS and STS).

Google Research Football offers two major mode of observations: 'simple115' and 'extracted' (also called the super mini-map).

The simple115 state representation is consists of coordinates of players, players' movement directions, the ball position, a ball movement direction, a one-hot encoding of ball ownership, a one-hot encoding of which player is active. This totals in a vector of length 115.

The extracted state representation consists 4 stacked layers of size $(72, 96)$. Layers contain one-hot encoding of spatial positions of game entities. These are (on the subsequent layers): players on the left team, players on the right team, the ball, and the active player.

We note that even though the extracted representation contains 'less information' than simple115, it has been reported in Kurach et al. (2019) to generate better results.

In our experiments, we use the so-called checkpoint rewards, which provide an additional signal for approaching the goal area. Details can be found in Kurach et al. (2019), where they were introduced and used in large-scale experiments.

The action space in GRF consists of 19 actions representing high-level football behaviors (e.g. "Short Pass"), see Kurach et al. (2019, Table 1).

Figure 7 shows training curves for our STS agent on Google Research Football. Training was run for 3 days until convergence. On the y-axis is the solved rate calculated as described above in Section A.8. On the x-axis is the number of real steps in the environment (planning steps in the simulator are not added). Curves are mean over 3 training runs with different seeds and shaded regions represent 95% confidence intervals. Moreover, to smooth the curves, data points are averaged in the windows of 10000 steps.

### A.8.1 SHOOTING METHODS

Tuning $c_{puct}$ turned out to be the most important one to make Bandit Shooting work, see Algorithm 4. In a nutshell, it needs to be adjusted to scale of rewards (value function) in a given environment. In our experiments we found $c_{puct} = 2.5$ to work best.

Using additional Dirichlet noise, $c_{noise} > 0$, and action sampling on the real environment, $\tau > 0$ (see Algorithm 7) resulted in inferior results with an exception of the "Counterattack hard" scenario.

### A.8.2 MCTS AND STS EXPERIMENTS

Apart from *multi-step expansion* we introduced another simple method, which might be of interest to the general public. Namely, before starting training, we set the weights of the last layer of the $Q$-value neural network to 0 (see Section A.3 for a detailed description of architectures). We observed that this significantly improved the training stability due to better exploration (and avoiding suboptimal strategies at the early stages of training). See 'No zero initialization' on Figure 12. This mechanism is similar to recent recommendations of (Andrychowicz et al., 2020, Section 3.2) given in the model-free setting.

## A.9 ABLATIONS AND ANALYSIS

This section is devoted to analysis of various aspects of STS and comparisons to MCTS.

### A.9.1 ANALYSIS OF THE TREE DEPTH

Recall that our hypothesis is that the benefits of *multi-step expansion* come from tilting the search towards DFS. While it is hard to formally prove this statement we were able to pin-point this effect in Sokoban and Google Football experiments, see Figure 8 and Figure 9 respectively.

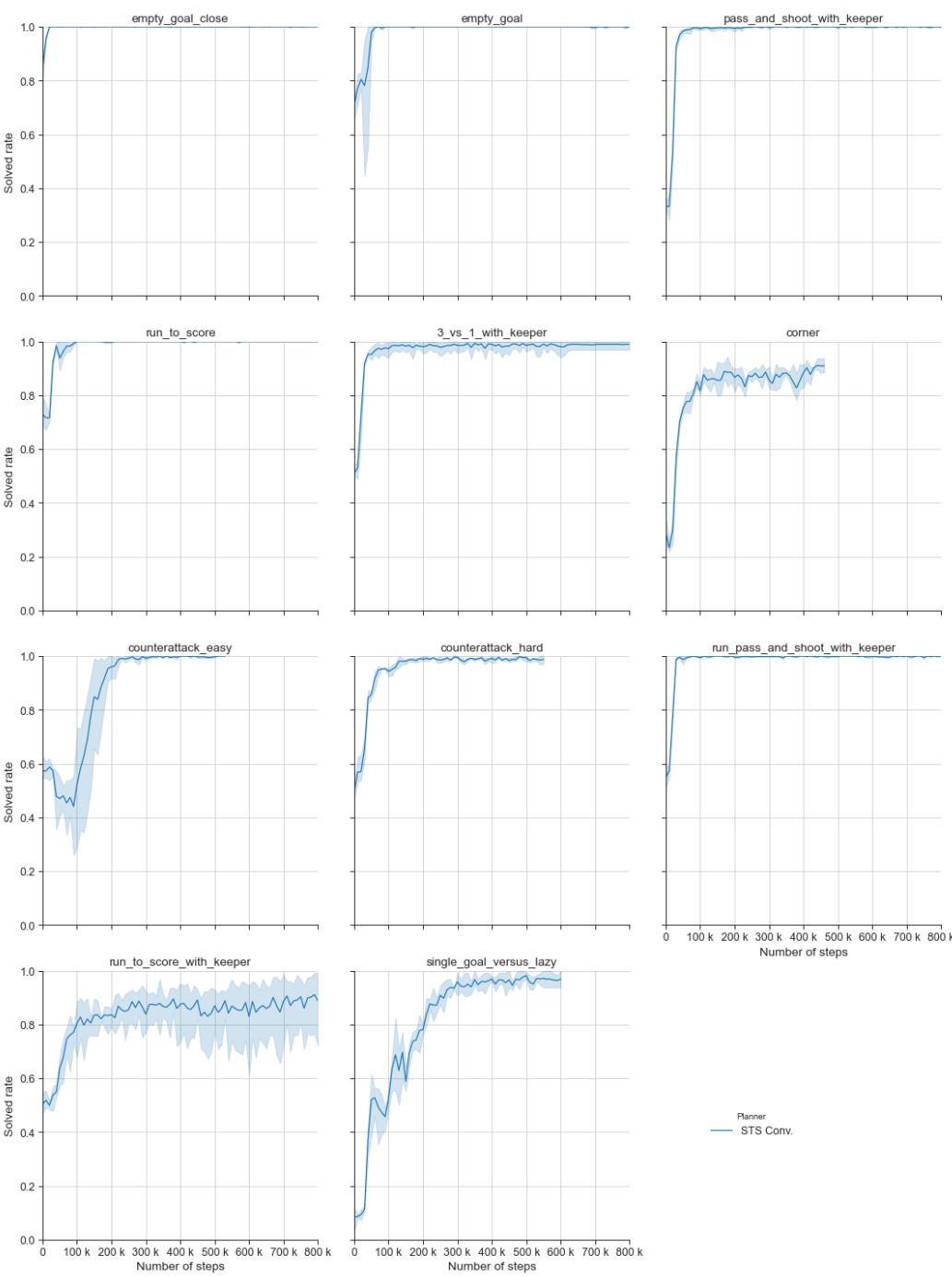

Figure 7: Google Research Football training curves for STS on GRF. Mean over 3 seeds with shaded regions representing 95% confidence intervals.

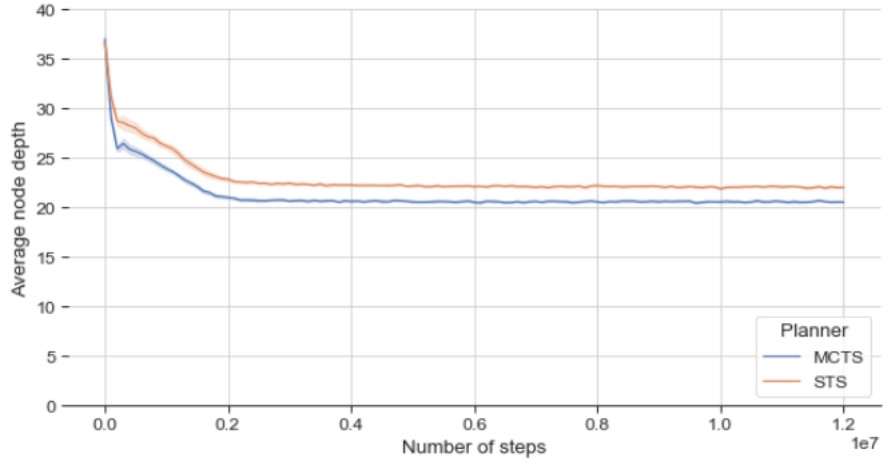

Figure 8: Search depth of MCTS and STS on Sokoban.

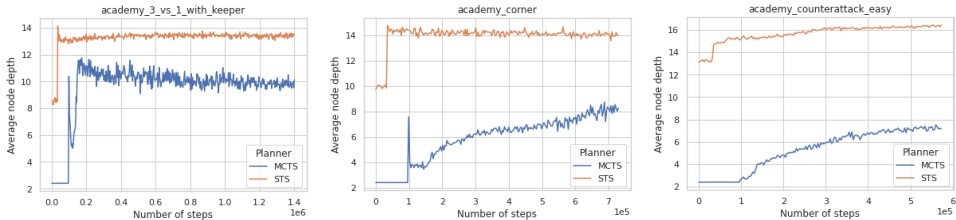

Figure 9: Search depth of MCTS and STS on three Football Academy tasks.

### A.9.2 COMPARISON TO ALPHAGO

We run two additional experiments with rollout-based evaluation using different policies. These are meant to provide more evidence to support our hypothesis that the benefit of STS is due to the *multi-step expansion* mechanism.

In each experiment, the rollout was truncated after 10 steps to ensure fair comparison with STS. The return after the last step of the rollout was approximated using the Q-value network. This value and rewards collected were used to calculate the leaf's value in the same way as in AlphaGo. In experiments, we tested two strategies for generating rollouts:

1. Actions sampled from the prior policy - the same setup as in AlphaGo, except for rollout truncation and the choice of the policy. AlphaGo used a policy pretrained on expert data. Since we do not have access to such data for Google Football, we instead used the prior policy trained over the course of the algorithm.

2. Actions chosen deterministically, to maximize $Q(s, a) + c_{PUCT} * \pi(a|s)$. $Q$ was computed by a neural network and $\pi$ is the probability given by the trained prior distribution. We recall that this setup is equivalent to STS except for the crucial fact that STS adds the expanded leaves to the search tree.

We observed that strategy 1. performed very poorly, which highlights the importance of using neural networks for leaf evaluation. Strategy 2. performed significantly better but still worse than STS. We believe that these results strengthen the evidence that the advantage of STS stems from the algorithmic reasons by building a more efficient search tree. Note the tasks on which the two evaluated strategies performed the worst, i.e. counterattack_easy, counterattack_hard, single_goal_versus_lazy, are those with the longest lengths of a successful episode. This supports the argument that STS better handles problems with long planning horizons.

| | Method | 3 vs. 1 with keeper | Corner | Counterattack easy | Counterattack hard | Empty goal | Empty goal close | Pass and shoot with keeper | Run pass and shoot with keeper | Run to score | Run to score with keeper | Single goal versus lazy |
|---|---|---|---|---|---|---|---|---|---|---|---|---|
| MCTS | Conv. | 0.81 | 0.50 | 0.31 | 0.31 | 0.99 | **1.00** | 0.45 | 0.89 | 0.70 | 0.00 | 0.00 |
| MCTS | Conv.+r.r. | 0.91 | 0.09 | 0.00 | 0.03 | 0.00 | **1.00** | **1.00** | **1.00** | 0.00 | 0.06 | 0.00 |
| MCTS | Conv.+d.r. | **1.00** | **0.81** | 0.06 | 0.35 | **1.00** | **1.00** | **1.00** | **1.00** | **1.00** | 0.94 | 0.25 |
| STS | Conv. | **1.00** | **0.81** | **1.00** | **1.00** | **1.00** | **1.00** | **1.00** | **1.00** | **1.00** | **0.97** | **0.97** |

Table 5: Comparison of STS with leaf evaluation using a Q-value network and MCTS with different leaf evaluations: Q-value network only (Conv.), network + deterministic rollout (Conv.+d.r.), network + random rollout using the prior policy (Conv.+r.r.). In all experiments we used the same convolutional network architecture. The reported results are the median solved rates across 3 runs.

### A.9.3 ABLATIONS OF STS DESIGN CHOICES

The ablations were performed on three environments from GRF Academy: *corner*, *counterattack hard* and *empty goal*, see Figure 12. The first two environments are difficult, while the last one is easy. The following parameters or settings were subject to analysis (they correspond to the labels in Figure 12):

- `prior noise weight`: a weight in the mixture of Dirichlet noise and the prior.
- `depth limit`: the maximum number of nodes visited in a single STS pass.
- `sampling temperature`: temperature for sampling the actions on the real environment.
- `MCTS n_passes 300`: this corresponds the standard MCTS setting with $H = 1$ (MCTS) and $C = 300$
- `Value network n_passes`: value network is used instead of $Q$-function. Note that `n_passes` $= 2$ matches roughly the $Q$-function version in terms of visited states (recall, see Section 5, that $Q$-function evaluates all children at once and that number of actions in GRF is 19).
- `No policy`: instead of a learned policy network, a uniform policy is used.
- `No zero initialization`: the last layer of the value function neural network was not initialized to 0 (see description at the beginning of Section A.8.2).

The default setup (denoted as `Prior noise weight 0.1`) is always positioned at the top in Figure 12. It uses parameters described in Table 3 in the Google Research Football STS column.

### A.9.4 ABLATION - BACKPROP WEIGHT

This ablation aims to verify if there is benefit of the aggregate backprop implemented in UPDATE in Algorithm 6. We compare to the method proposed by Soemers et al. (2016), which backpropagates only the last value. We observed a clear advantage of STS on the Sokoban domain, see Figure A.9.4.

### A.10 MULTI-STEP EXPANSION ANALYSIS ON TOY PROBLEMS

First, consider an MDP presented at the top of Figure 11. It showcases the situation when the errors are systematic: in the vicinity of the starting state $s_0$, the estimates of the value function are biased (for simplicity set to 0 and shown as white vertices), while the values in the area surrounding terminal states are accurate (shown as color vertices). This example is an exaggeration. However, something similar can happen in practice, when information is propagated with $TD$-like methods or the environment has an "easy" region, which is hard to find. Under these circumstances, STS, given large enough $H$, will be able to reach accurate values (color vertices) within a few passes. On the contrary, MCTS would explore the whole uncertain area (white vertices) in a breadth-first fashion.

Second, consider an MDP shown at the bottom of Figure 11. It illustrates the case when the errors are "pseudo-random". In this MDP all rewards are 0 except the marked edges, where they are $-a, a > 0$.

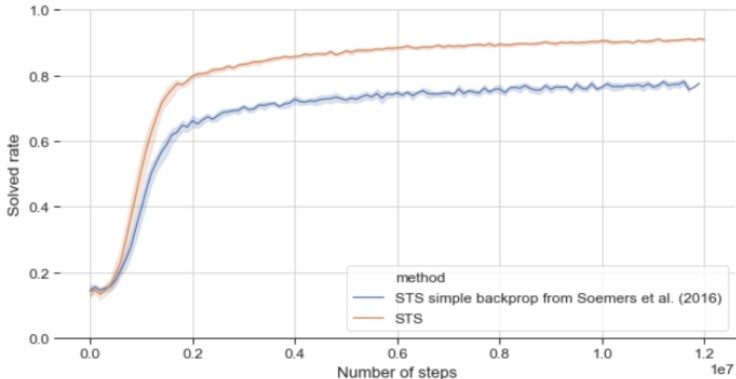

Starting from $s_0$, the agent can move only to the right. The perfect value function is $0$ in every state, however we assume that the current noisy value estimates equal to $\epsilon_i$ on the "tail" part of the diagram. In this example, we assume that the errors arise in interactions of many factors, thus can be modeled as i.i.d. centered random variables $\epsilon_i$ such that $\mathbb{E}|\epsilon_i| < +\infty$.

The optimal path, going over the green edge and later over the tail, is accompanied by several 'decoy' paths (marked in orange). They will not be entered unless errors on the tail have accumulated below $-a$. We denote the probability of such an event by $p_H$, where $H$ is the number of steps in the multi-step expansion ($H = 1$ corresponds to MCTS). In Lemma A.10.1, we show that $p_1 > p_H$ for $H \geq 2$, and in fact $p_H \to 0$ when $H \to +\infty$.

**Lemma A.10.1.** *Under the above assumptions $p_1 > p_H$ and $p_H \to 0$.*

*Proof.* Assume that for the first $\ell \geq 2$ steps of the search tree was unfolded via the middle (green) edge and further via the tail. The state-action value estimated by the MCTS/STS is thus $q_\ell = (\epsilon_0 + \ldots + \epsilon_{\ell-2})/\ell$. Consequently,

$$p_H = \mathbb{P}(\exists_{k \in \mathbb{N}} q_{kH} < -a).$$

The claims follow from the fact $q_\ell \to 0$ a.s., which itself is the consequence of the strong law of large numbers. $\square$

As the lemma serves mainly the illustrative purpose we used the i.i.d. assumption, which can be easily weakened. As a test we simulate the case $\epsilon_i \sim \mathcal{N}(0,1)$ and $a = 0.3$. In this case $p_1 = 0.56, p_2 = 0.46, p_4 = 0.35, p_8 = 0.41, p_{16} = 0.21$. Note that $p_1/p_H$ is as high as $3$ for $H = 16$ and quite natural choice of $a$ and $\epsilon_i$.

### A.10.1 HIGH COMPUTATIONAL BUDGETS

STS is predominantly meant to improve the search for modest computational budgets. When the budget gets bigger, any search becomes more exhaustive, and the benefits are likely to diminish. To our pleasant surprise, see Figure 10, we still observe certain advantages of STS in later phases of training, even though MCTS behaves better initially. This observation might suggest another interesting line of inquiry, namely, developing methods for adaptive breadth/depth balancing (e.g. via changing $H$).

### A.11 INFRASTRUCTURE USED

We ran our experiments on clusters with servers typically equipped with $24$ or $28$ CPU cores and 64GB of memory. A typical experiment was $72$ hours long (the timeout set on the clusters), which was enough for most experiments. Experiments that did not converge during this time were resumed.

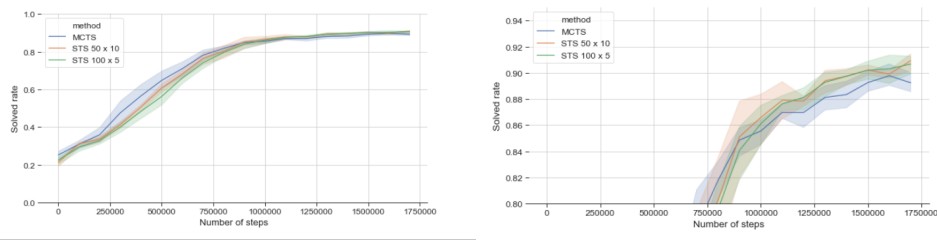

Figure 10: Experiment with big computational budget $C \cdot h = 500$ on Sokoban.

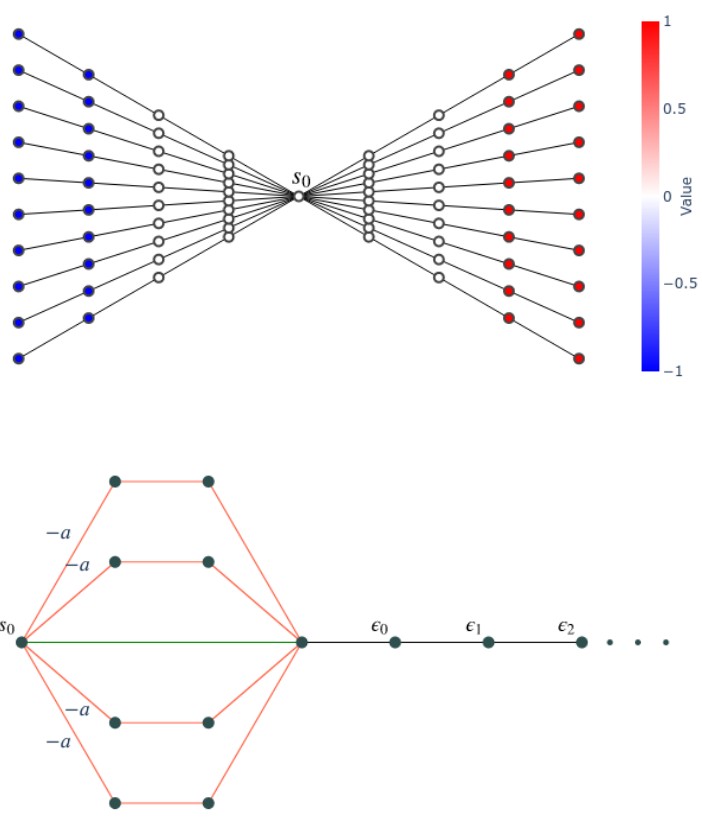

Figure 11: Visualization of the toy environments.

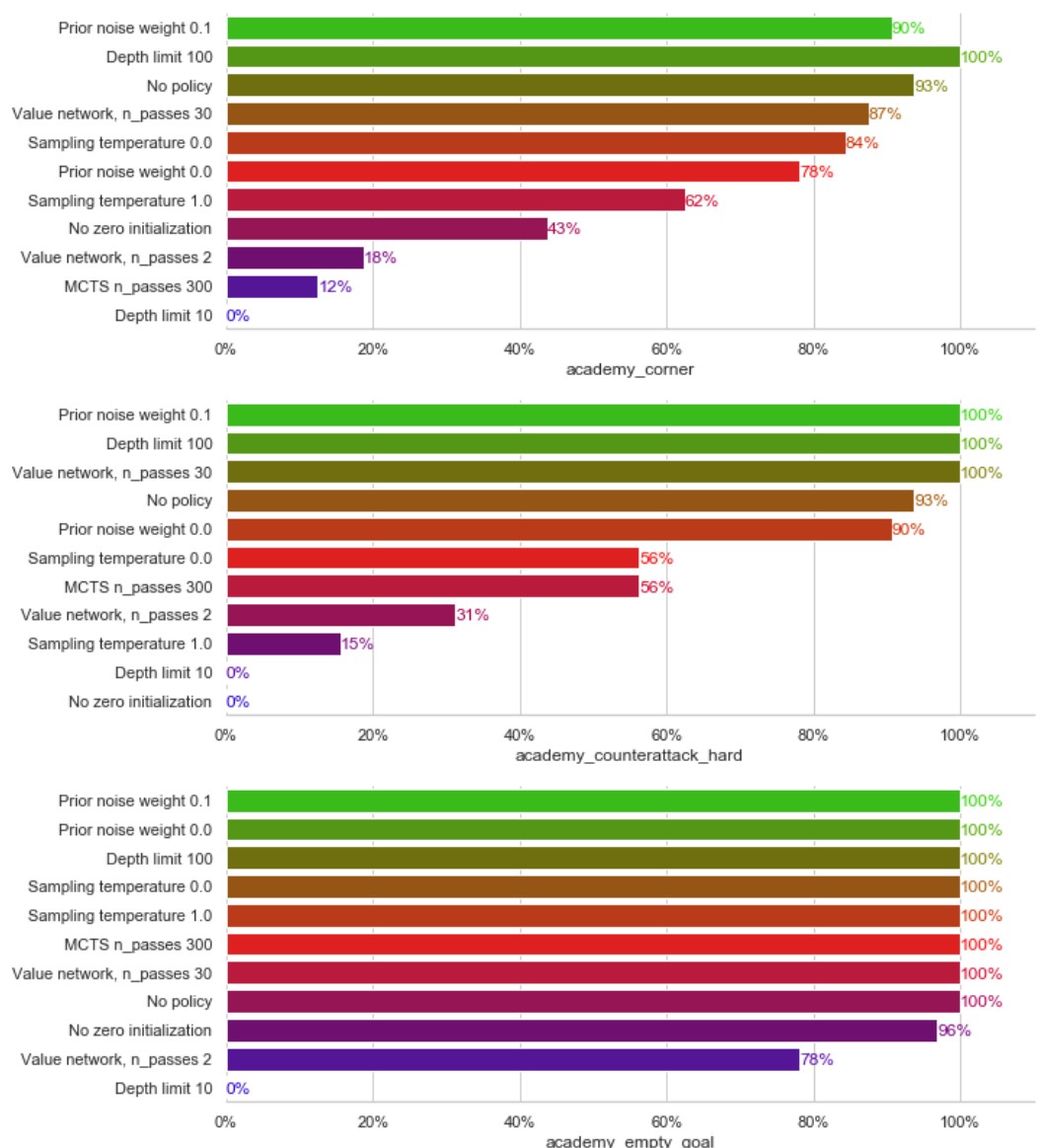

Figure 12: Ablations performed GRF Academy environments: *corner*, *counterattack hard* and *empty goal*.

