# OpenReview forum: "Structure and randomness in planning and reinforcement learning"
_ICLR.cc/2021/Conference — Reject_

### Official Review · AnonReviewer1 · 2020-10-25

**Rating:** 6
**Confidence:** 4

**Review:**

Summary:
---

The paper presents "Shoot Tree Search", an approach that can basically be summarised as a variant of MCTS that expands (adds to the search tree) a longer sequence of up to H nodes to the tree per iteration, as opposed to the standard approach of expanding a single node per iteration. The experiments demonstrate improved performance in comparison to a "standard" MCTS and a variety of simpler rollout-based planning approaches, in challenging planning domains such as Sokoban and Google Research Football.

Strong Points
---

1) Well-written, mostly easy to read and understand.
2) Simple but interesting idea.
3) Thorough empirical evaluation, interesting results.

Weak Points
---

1. The paper describes the modification of MCTS into STS, which consists of making it expand a longer sequence of up to H nodes within a single iteration, as an entirely novel way to extend MCTS, but I'm not sure that that's entirely the case.  For instance, Coulom's 2006/2007 paper "Efficient Selectivity and Backup Operators in Monte-Carlo Tree Search" already states: "In practice, not all the nodes are stored. Storing the whole tree would waste too much time and memory. Only nodes close to the root are memorized.", which suggests that something like this may have already been considered, but in that case was not found to be worthwhile. The 2016 paper "Enhancements for Real-Time Monte-Carlo Tree Search in General Video Game Playing" describes "In this paper, the tree is simply expanded by adding the whole play-out to the tree.", which seems similar.
I do still like that the paper performs a thorough evaluation of this idea, which I am not aware of appearing in previous literature, and the setting with DNNs for value / policy function approximations is also different from aforementioned papers which may lead to different trade-offs. The use of a DNNs for value function probably changes the story quite a bit here, because the longer horizon H also changes the point at which the value function is computed, as opposed to those older papers with values estimated by random rollouts (which remains the same regardless of the horizon H). So I'm not saying the idea isn't "novel enough", just that some discussion of past work seems to be missing.

2. In my experience, the primary reasons historically for the typical strategy of expanding just 1 node per iteration in standard MCTS (without DNNs) are 1) to reduce memory usage (especially when copies of game states are stored inside nodes, because then every node can be quite big), and 2) efficiency, because if you store copies of game states in nodes, and create more nodes, you also need to copy more game states (whereas a random playout without node and state storing can just roll out at once without making intermediate copies of states). I'm kind of missing a discussion of these kinds of considerations.

3. I'm not sure that I can fully understand the experiment setup, in particular looking at Table 1. C is a hyperparameter denoting the number of planning passes, and N_p is described as "the average number of passes until the solution is found". How can N_p ever exceed C? Shouldn't it be upper bounded by C? I guess C might be the number of planning passes "per time step", and N_p is total over the entire episode, something like that? But this is not really clear to me. If the algorithms are really restricted to just C iterations of MCTS, I guess it's fair to always keep C*H constant and then my points above about memory usage / efficiency are not a big deal since they would still be equal across all scenarios... but I'm a bit confused here due to N_p exceeding C.

Overall Recommendation
---

Right now I have too many little points of confusion / missing discussion, as pointed out under "weak points" above, to recommend acceptance. That said, there is also enough to like about the paper, and I can easily envision that most of the points of confusion could be relatively straightforward to clear up in a revision.

Questions for authors
---

Could you please clarify on the points raised under "weak points" above?

Minor Comments
---

- On first page, the comma in "Google Research Football is, an advanced" seems unnecessary and confusing.
- On page 6, the wording "Shooting methods perform poorly for Sokoban" could be confusing because the newly proposed "Shoot Tree Search" method can very easily be interpreted as also being a "shooting method" due to its name.
- In Lemma A.6.1, the assumption that STS and MCTS build the same tree T seems to me like it's a VERY strong assumption; the MCTS has to make very very specific choices, with very frequent overlap making identical choices across different iterations (inherently somewhat unlikely due to the visit count terms in PUCT and other Selection strategies), for this to be true.

After Discussion
---

I increased my review from marginally below to marginally above acceptance threshold. Most of the remarks I had were at least partially addressed. If the paper gets accepted, I'd still recommend looking at some of them again and clarifying more. A simple, explicit remark somewhere around Table 1 explaining that N_p can indeed exceed C due to relevant parts of the search tree being preserved across time steps would help a lot. Some more explicit discussion about why the difference between using a trained value functions vs. heuristics / terminal results matters so much that it makes this substantially different from prior work would also help (I understand that it is because in prior work the only advantage of storing all those extra nodes was really just that it could retain slightly more information from backpropgations in those nodes, whereas in your case it changes which state is the state that gets evaluated by a trained value function, but this should be more explicit in the paper).

---

> ### Author Response · Authors · 2020-11-17
> **We thank the reviewer for the detailed review. We believe that it will lead to the improvement of our work; we are preparing the revision**
>
> We thank the reviewer for the detailed review. We believe that it will lead to the improvement of our work; we are preparing the revision.
>
> **Weak point 3**: We agree that this part of the text could be written more clearly, and we will do so in the revised version. Answering your question, you are right: C is the number of passes in one planning step, while N_p is the total number of passes in the whole episode (until the solution is found). We also point to Table 4 (extending Table 1) to give more intuition about the memory used. One more clarification is perhaps also worth stating. In every planning step, an action is chosen, and the subtree corresponding to this action is retained to the next planning step. This is a rather standard: it improves the search quality but also increases memory consumption. In principle, the latter could be problematic, however, we did not observe this to be the case in our experiments.
>
> **Weak points 1 and 2**: we thank you for pointing out the references [1] and [2], we will include them in the related work section with appropriate discussion (see the text below). Our approach is different, in the sense that we operate in the modern 'post-AlphaZero' context. We are interested in algorithmic aspects, and our work is meant to make some steps towards understanding the tradeoff between breadth-first search and depth-first search as well as between bias and variance. As far as we understand, this is not present (explicitly) in these previous works. More precisely, let us point out the differences with [1]. Each STS step is composed of three elements: a) expansion of H consecutive nodes, b) addition of the expanded nodes to the tree, c) evaluation of H expanded nodes by a neural network value function approximator and backpropagation of each of these values (which for better efficiency is squashed in one 'mega' backpropagation step; hence the code for UPDATE in Algorithm 6). All of this is embedded in a reinforcement learning training loop. Although [1] expands multiple nodes, it backpropagates the 'game score value' of a final state of the simulation. We on the other hand learn this value (using RL) and benefit from the averaging effect of multi-step expansion. In the course of research leading to this publication, we performed experiments with multiple backpropagation schemes on Sokoban, which underperformed, being even worse than the standard MCTS. This ablation will be included in the revision.
>
> We also note that, if properly used, our method does not increase memory usage much. We observed this throughout many experiments and checked rigorously in Sokoban's isolated setting, see Table 1 and Table 4. We argue that for moderate values of the multi-step expansion, the size of the tree does not increase significantly, and sometimes even decreases. In our view, this supports the hypothesis that STS builds 'a better search tree', when H is appropriately set. From our experiments, we recommend H=10 as the starting point.
>
> Concerning [2], the work presents interesting ideas of adding rollout (or their parts) to the search tree and various backup operators (including uncertainty awareness). This is similar to STS, though the crucial difference is that we use neural network estimates of values and never do full rollouts.
>
> We also thank you for ‘minor points’ which we agree with. We will modify the text accordingly. In particular, Lemma A.6.1 is meant to be merely an illustration. Having understood that the assumption is indeed strong, we decided to put it only into the appendix (we consider removing it in the revision).
>
> [1] Enhancements for Real-Time Monte-Carlo Tree Search in General Video Game Playing
> [2] Efficient Selectivity and Backup Operators in Monte-Carlo Tree Search

---

> ### Author Response · Authors · 2020-11-20
> **New experimental results**
>
> We gently note that we update our answer with some new experimental results in our answers to the other reviewers.

---

### Official Review · AnonReviewer2 · 2020-10-30
**A modification of Monte Carlo tree search that produces marginal improvements that may not be present with tuning of the Monte Carlo tree search exploration parameter**

**Rating:** 3
**Confidence:** 4

**Review:**

The authors present a method that combines Monte Carlo tree search (MCTS) and random rollouts. The authors their relate this to the bias-variance tradeoff observed in n-step temporal difference methods. The authors evaluate their method on Sokoban and the Google Football League environment. The results show that the authors' method leads to marginal improvements on these domains.

I do not think what the authors are doing is very novel as MCTS combined with rollouts was already used in AlphaGo. Furthermore, I believe the small difference in results can be made up by using only MCTS with a different exploration parameter (i.e. like the one that was used in the AlphaGo paper).

I would like to know what benefits this method brings that cannot be obtained from combining MCTS with rollouts as in AlphaGo or from a hyperaparameter search with MCTS. Is there an anaylsis of the bias variance tradeoff of this method?

---

> ### Author Response · Authors · 2020-11-17
> **Thank you for your review**
>
> Thank you for your review.
>
> There is an important difference between STS and the variant of MCTS used in AlphaGo - STS adds states visited during the rollout from the leaf to the tree, while AlphaGo does not (we currently run and going to add an ablation comparing STS to AlphaGo-style simulation to the paper). This has several implications: a) due to the averaging, a better value is backpropagated up the tree (technically, this is more akin to TD-lambda as we evaluated the value function estimate in each step of multi-step expansion), b) arguably a more efficient search tree is built. Our intuition is that the tree is deeper, and the paths explored in multi-step expansion can easily be branched out during later planning passes.
>
> On the experimental side, we provide an isolated study in Table 1, where some effects on the tree statistics can be seen. Last but not least, Table 2 shows substantial improvements in GRF (e.g. 40%-100% absolute improvement in solved rate on the more challenging tasks).
>
> We run several thousands of tuning experiments (including a search over the PUCT parameter), which leads us to believe that the mentioned performance improvement is due to algorithmic properties of the proposed STS mechanism. We note that we concentrate on rather modest computational budgets, as in our view, it is an important regime for applications. We speculate that in this regime, algorithmic improvements might be more relevant as opposed to more brute-forced cases.

---

> ### Author Response · Authors · 2020-11-20
> **We update our answer with new experimental results**
>
> We update our answer with new experimental results. Here we present the most relevant results to this review and we encourage the reviewer to take a look at the remaining answers, where we also discuss the results of several new experiments.
> We thank the reviewer for the suggestion for comparison with AlphaGo-style leaf evaluation using rollouts. We have run two additional experiments with rollout-based evaluation using different policies. In each experiment, the rollout was truncated after 10 steps (to complete them before the end of the rebuttal phase and ensure fair comparison with STS).
>
> The return after the last step of the rollout was approximated using the value network. This value and rewards collected were used to calculate the leaf's value in the same way as in AlphaGo. In experiments, we tested two strategies for generating rollouts:
>
> 1. Actions sampled from the prior policy - the same setup as in AlphaGo, except for rollout truncation and the policy's choice. AlphaGo used a policy pretrained on expert data. Since we do not have access to such data for Google Football, we instead used the prior policy trained over the course of the algorithm.
> 2. Actions chosen deterministically, to maximize Q(s, a) + exploration_weight * \pi(a | s). Q was computed by a neural network and \pi is the probability given by the trained prior distribution. We recall that this setup is equivalent to STS except for the crucial fact that STS adds the expanded leaves to the search tree.
>
> We observed that strategy 1. performed very poorly, which highlights the importance of using neural networks for leaf evaluation. Strategy 2. performed significantly better but still worse than STS. In our opinion, these results strengthen the evidence that the advantage of STS stems from the algorithmic reasons by building a more efficient search tree. Note the tasks on which the two evaluated strategies performed the worst, i.e. counterattack_easy, counterattack_hard, single_goal_versus_lazy, are those with the longest lengths of a successful episode. This supports the argument that STS better handles problems with long planning horizons.
>
> (In experiments, we used three seeds and we reported their median. Numerical results are available [here](https://postimg.cc/94GjXdrH) 1 is marked as (Q + policy r.) and 2 as (Q + det. r.)).

---

### Official Review · AnonReviewer4 · 2020-10-30
**The paper presents a simple extension to MCTS search by choosing multiple actions in each call to 'expansion' phase. The main concern with the paper is the number of simulations for MCTS.**

**Rating:** 6
**Confidence:** 2

**Review:**

**Summary**
This paper presents a new planning algorithm, called Shoot Tree Search, to control the trade-off between depth and breath of the search. STS modifies the expansion phase of tree search by choosing multiple actions (e.g. $\gt$ 1) instead of one level expansion. The presented idea is simple and straightforward and seems to provide improvement over existing tree-based planning algorithms. The presented detailed ablation studies provides insights about the choices made in the paper.

**Reasons for score**
Overall, I liked the paper and the simplicity of the idea. However, my major concern is the comparison with MCTS. I am not convinced that STS would outperform vanilla MCTS when the number of simulations is in order of thousands (e.g. the number of simulations in AlphaGo paper is around 1600).

**Strengths**
+ The idea is simple and seems to outperform vanilla MCTS implementation in the environments with large action space.

**Weaknesses**
+ The comparison with the related work is not thorough which makes it hard to come into a decisive conclusion about the performance of the proposed method.
+  There are some missing related work, e.g. using policy network for multiple rounds of simulations.

**Questions**
+ What would the benefits if we have a policy network to perform the rollouts (e.g. a similar method to [1])?
+ In general, the benefit of MCTS algorithm (like AlphaGo which performs around 1600 simulations) presents itself when the number of simulations are large. Can you compare running MCTS with more number of simulations (e.g. large C) and STS?
+ Can you please provide some insights on why in 'Corner' STS underperform compared to random shooting?

[1] https://cs.brown.edu/people/gdk/pubs/analysis_mcts.pdf

---

> ### Author Response · Authors · 2020-11-17
> **We thank the reviewer for comments and questions. We will prepare a revised version of the paper, which in particular will expand on the related work**
>
> We thank the reviewer for comments and questions. We will prepare a revised version of the paper, which in particular will expand on the related work.
>
> We agree that the difference between MCTS and STS might be negligible when the number of simulations is large. That being said, we see much value in developing methods that perform well with smaller computational budgets. There is a clear practical aspect: using large numbers of passes (like aforementioned 1600) makes the method out of reach for many real-world uses, where the model of the environment is costly to run. Operationally, studying the methods relying on high computation budgets is the luxury that only several big and well-funded research labs can afford. From a more theoretical (or philosophical) point of view, we argue that putting constraints on the computational budget might be an important aspect of measuring 'intelligence'.  Although we are fine with the fact that many recent advancements in AI heavily hinged on computational power, we sympathize with the view that the learning system's quality should also be measured along the resources axis. As neatly phrased by Lake et al. [1]: "One worthy goal would be to build an AI system that beats a world-class player with the amount and kind of training human champions receive – rather than overpowering them with Google-scale computational resources."
>
> Answering questions:
> 1. This is an interesting question. We believe that the series of papers [3], [4], [5] provided quite substantial evidence that the MCTS planner with value function evaluation (AlphaZero) replacing the policy rollouts (AlphaGo) is more powerful and simpler. Using rollout policies has several disadvantages. Some can be exemplified by the environments used in our experiments. In Sokoban, for instance, there are multiple ‘dead-end’ states (i.e., states from which the agent cannot reach the goal position) and our experiments showed that planning is essential for avoiding these pitfalls. A pure neural network performs much weaker (usually a drop is around >20% of solved ratio), and even worse for a random policy rollout. In GRF, using rollout is perhaps a better option, although it comes at the cost of computation needed to run this complex simulator.
> We also note that ‘shooting experiments’ can be seen as a proxy to answer this question. A slightly speculative conclusion would be that for some environments using rollouts yields much worse results (due to bias and variance) - Sokoban in our case.  There are also environments where the estimates are sharp enough to get progress (GRF in our case and environments in [2]). Understanding the circumstances when such property holds is an interesting research question.
> 2. As mentioned above, we concentrate on a modest computation regime due to conscious philosophical and practical choices. However, the question asked is a valid one; we currently run an experiment with a bigger number of passes and will update the answer once it is done.
> 3. It is a good question. We speculate that this is an exploration issue as due to the particular construction of rewards in GRF, the reward in the corner scenario is more sparse (the so-called checkpoint rewards are not available). Moreover, random rollouts can still provide a reliable evaluation, which probably adds up to the ‘shooting’ victory. Note, however, that the STS result is also quite high, and well above the results of the remaining methods. Having said that, more investigation is needed to provide a definite answer.
>
>
> [1] Building Machines That Learn and Think Like People, Lake at al.
> [2] https://cs.brown.edu/people/gdk/pubs/analysis_mcts.pdf
> [3] Mastering the game of Go with deep neural networks and tree search - Silver, D. et al. 2016.
> [4] Mastering the game of Go without human knowledge -  Silver, D. et al. 2017.
> [5] A general reinforcement learning algorithm that masters chess, shogi, and Go through self-play. Silver, D. et al. 2018.

---

> ### Author Response · Authors · 2020-11-20
> **We update our answer with new experimental results**
>
> We update our answer with new experimental results. Here we present the most relevant results to this review and we encourage the reviewer to take a look at the remaining answers, where we also discuss the results of several new experiments.
>
> We present partial results with a bigger number of passes. A complete experiment will be presented in the final version of the paper.
>
> We ran a Sokoban experiment with an expansion of 500 nodes per move. We found out that after 1 million steps STS and MCTS had similar results (MCTS 85.5% solved rate, STS 86/86.5% *). It is hard to draw definite conclusions; we speculate that MCTS gains some advantage in early training due to more methodical BFS-like search (possible within a high computational budget). These gains seem to disappear later on, suggesting that STS works well also in large scale settings.
>
> (We ran two versions of STS - (1) 100 passes, H=5 and (2) 50 passes, H=10. Results are averaged over 5 runs. Graphs can be found [here](https://postimg.cc/4KrTs8J5)).

---

### Official Review · AnonReviewer3 · 2020-11-02
**New MCTS algorithm for large state spaces**

**Rating:** 4
**Confidence:** 3

**Review:**

Summary:
This paper proposes a new algorithm named ‘Shoot Tree Search (STS)’ to perform planning in large state spaces. The authors construct STS by redesigning the expansion phase of MCTS using multi-step expansion. The authors provide pseudocode of the STS and compare the performance of STS and MCTS empirically in various domains, such as Sokoban, Google Research Football (GRF).
Comments:
Firstly, there is no intuitive explanation of why, what and how. Even after reading the paper, I do not agree that STS is good, because there is no intuition as to why it is better than naïve MCTS. More detail, I have a question - The main difference between STS and MCTS seems to be using multi-step expansion or 1-step expansion. Although multi-step expansion will gather more information about (s,a) pairs with high Q(s,a) value (because the actions chosen by argmax Q and STS expands such trajectories), but in sparse reward problem, STS and MCTS will work similarly. Moreover, before getting positive reward, STS may worse than MCTS because it requires more samples to explore (because STS uses more samples for (s,a) pairs with high Q-values, which is not meaningful yet). So I think that this paper needs at least discussion on an intuitive level about the advantage of STS.
In addition, the empirical details in appendix (figure 7 and 8 on page 18 and 19, respectively) look weird – each algorithm seems to have stopped randomly or incompletely.
Also, the authors seem to need to make an effort to make the paper more self-contained.
Minor comments
Some abbreviations are used without its full word or phrase. For examples, MCTS (it has been used in page 1, but the full phrase appears on page 3), and RL.
There are no reference for random shooting and bandit shooting. The authors should provide more explanation about them with references.

---

> ### Author Response · Authors · 2020-11-17
> **Thank you for your review. We will prepare a revised version of the paper that takes the reviewer's comments into account, in particular putting more emphasis on intuition in the paper and arranging the text so it can be found in one place, as well as improving presentation of some of the results**
>
> Thank you for your review. We will prepare a revised version of the paper that takes the reviewer's comments into account, in particular putting more emphasis on intuition in the paper and arranging the text so it can be found in one place, as well as improving presentation of some of the results (e.g. figure 7 and 8).
>
> Intuitively, STS enables building the search tree, which might be more efficient. We found it particularly relevant in Google Football, where individual actions induce rather small changes in the environment. Experimentally, we found that MCTS builds a relatively wide tree, which poorly explores the state space. This observation is also confirmed in isolated experiments presented in Table 1 (Table 4). STS has a smoothing effect, reducing biases in neural-net value estimators. Having said that, we would like to draw the Reviewer’s attention to the following parts of the paper:
> * The introduction describes that STS can be viewed as a mechanism controlling depth and breadth of the search and can be viewed as a bias-variance control method, hence giving STS characteristics of interpolation between MCTS and random shooting.
> * Section 4.1 mentions STS ability to exit from the erroneous region more quickly by correcting biased value functions estimates.
> * In Section 4.2, we provide evidence that STS gives a boost in environments requiring long-horizon planning.
> * We devote Appendix A.7.3. to dig deeper into intuition concerning the role of STS in reducing bias.
>
> We believe the case of sparse rewards to be rather orthogonal to the relative performance of MCTS and STS. One could even argue that STS might perform better by introducing more directed exploration in the form of longer “shots” that have a higher chance of reaching the goal state than the “wide” exploration induced by MCTS. At the moment, it is just a highly hypothetical claim that requires an experimental verification (we run a simple experiment in a sparse version of Sokoban). To deal with sparsity, additional methods are required, regardless of the fact whether MCTS or STS is used. We expect that STS would blend smoothly with most of such methods. We leave this research question for a new project.

---

> ### Author Response · Authors · 2020-11-20
> **Update with new experimental results**
>
> Here we present the most relevant results to this review and we encourage the reviewer to take a look at the remaining answers, where we also discuss the results of several new experiments.
> We conducted experiments with STS and MCTS with the sparse reward version of Sokoban. Namely, reward is obtained only for solving the board. We have not observed significant differences from the previous experiments; in particular, STS is visibly better than MCTS. We speculate that the setup presented originally in the paper is already quite sparse (additional reward is presented by placing the first of two boxes).
> (Results are available [here](https://postimg.cc/DJM4v3SN); results are averaged over 5 runs for sparse settings and 10 runs for dense)

---

### Official Review · AnonReviewer5 · 2020-11-06
**rather weak paper**

**Rating:** 3
**Confidence:** 4

**Review:**

summary:
This paper introduces Shoot Tree Search (STS), a planning algorithm that performs a multi-step expansion in Monte-Carlo Tree Search. Standard MCTS algorithms expand the search tree by adding one node to the tree for each simulation. In contrast, the proposed STS adds multiple nodes to the search tree at each simulation, where each node corresponds to the state and action that are encountered during rollout. By multi-step expansion, the evaluation of the trajectory is less-biased, which can be analogous to n-step TD. In the experiments on Sokoban and Google research football domains, STS outperforms baselines that include Random shooting, Banding shooting, and MCTS.


Overall, my main concerns are technical novelty and presentation quality.

The most common MCTS methods assume that the leaf node is expanded one at a time in each simulation (and its evaluation is performed either by rollout policy or by function approximator), but this common practice does not necessarily mean that MCTS should always do that. The main reason for only expanding one node per simulation in standard MCTS is memory efficiency: if we fully expand the rollout trajectory and retain its information to the search tree, we may get slightly more accurate value estimates. However, the nodes located deep in the tree will not be visited more than once in most cases, thus its effect is usually not significant, leading to the common practice of one-step expansion. More importantly, multi-step expansion has already been used in existing works (e.g. in [1], the tree is expanded by adding the whole rollout trajectory), thus I am not convinced that this work introduces a technical novelty.

It seems that the relative benefit of the STS over MCTS observed in the experiments comes from the bias of the value function approximator. However, to show the effectiveness of 'multi-step' expansion compared to 'single-step' expansion, I think that more thorough ablation experiments should have been conducted. For example, we can consider the setting where both STS and MCTS perform leaf-node evaluation (i.e. UPDATE in Algorithm 5) by executing rollout policy rather than by using value function approximator. By doing so, we can focus only on the benefits of STS's retaining information of full rollout trajectory (i.e. multi-step expansion), compared to MCTS's retaining one-step information (i.e. single-step expansion) while eliminating the effect of biased value function estimation.
To relieve too much bias in the current MCTS's leaf node evaluation, mixing MC return of rollout policy and the output of the value network could also have been considered, as in AlphaGo (Silver et al. 2016). It would be great to see if STS still has advantages over MCTS in various leaf node evaluation situations.

Also, more writing effort may be required, and the current version of the manuscript seems premature to be published. There are some unclear or questionable parts.
- Algorithm 3 and Algorithm 4 are not the contributions of this work, thus they can be removed or moved to the Appendix. Instead, more discussions regarding the proposed method should have been placed in the main text.
- In Algorithm 2: the definition of CALCULATE_TARGET is missing.
- In Algorithm 5: In SELECT, the tree policy is defined by CHOOSE_ACTION that selects purely greedy action. If this describes the MCTS used in the experiments, I would say this is wrong. To make MCTS be properly working, an in-tree policy that balances exploration vs. exploitation is required (e.g. a classical choice is UCB rule).
- In Algorithm 6: In UPDATE, $N(s,a)$ and $quality$ are increased by $c$ times more, which means that the longer rollout length, the more weight is given. What is the reason for assigning more weight to the trajectory that has a longer rollout length? If the entire planning horizon is limited to finite length, this means that early simulations (short $path$ length, long $rollout$ length) have more weight than later simulations (long $path$ length, short $rollout$ length), but I do not think this is desirable. Is my understanding correct?
- For the Sokoban experiments, the pre-trained value function would significantly affect the performance of MCTS and STS, but I could not find the way how the value function was pre-trained.
- In Appendix A.2., the hyperparameters for Shooting and STS are very much different. Why did you set Shooting's hyperparameter differently from STS (e.g. VF zero-initialization, action sampling temp, etc.)?
- It seems that the choice of zero-initialization of the value network is rather arbitrary. I am not convinced that this would always work better. In some situations, optimistic initialization of the value network may be helpful to encourage exploration of the uncertain state regions.
- In Table 2, Why does RandomShooting-PPO underperform PPO? Since RandomShooting-PPO puts additional search efforts upon PPO, I expected that RandomShooting-PPO must outperform PPO.
- Table 5 could have been moved to the main text, replacing Table 2.

[1] Soemers et al., Enhancements for Real-Time Monte-Carlo Tree Search in General Video Game Playing, 2016 IEEE Conference on Computational Intelligence and Games (CIG 2016)

---

> ### Author Response · Authors · 2020-11-17
> **Thank you for the detailed review - it will improve the quality of our work. We will release a revision of the text and new experimental results. In what follows we provide a detailed answer.**
>
> We perceive the main benefit of our method in the fact that it builds a search tree of a different shape. Put differently, STS expands the tree with `macro-actions` consisting of H consecutive steps. We found it instrumental in the Google Football experiments, in which a typical action has a small effect on the environment state, and thus also the value of that state. As a result, standard MCTS struggles by falling into the “breadth-first” type of search; see Table 5 for full results. This causes STS to perform significantly better within the same computational budget. We understand the concerns regarding the increased memory usage. This could indeed happen if the length of the multi-step expansion was set to high. However, for moderate values (we typically use H=10), it does not seem to be the case. This is confirmed in synthetic experiments on Sokoban; see column $N_t$ in Table 1. Interestingly, in some cases, the STS search tree is slightly smaller than the MCTS one.
>
> It might be worth highlighting that in all experiments (including the one referred to above), we ensure a fair comparison of MCTS and STS by providing the same computational budget. More precisely, we set the number of passes $C_{MCTS}$, $C_{STS}$ and the multi-step expansion depth H so that $C_{MCTS} = C_{STS} \cdot H$ (see Algorithm 1 for notation).
>
> We thank the reviewer for bringing [Soemers et al., 2016] to our attention; we shall include this work in the revision. Let us, however, point out the differences. Each STS step is composed of three elements: a) expansion of H consecutive nodes, b) addition of the expanded nodes to the tree, c) evaluation of H expanded nodes by a neural network value function approximator and backpropagation of each of these values (which for better efficiency is squashed in one ‘mega’ back-propagation step; hence the code for UPDATE in Algorithm 6). All of this is embedded in a reinforcement learning training loop. Although [Soemers et al., 2016] expands multiple nodes, it backpropagates the 'game score value' of a final state of the simulation. We on the other hand learn this value (using RL) and benefit from the averaging effect of multi-step expansion. In the course of research leading to this publication, we performed experiments with multiple backpropagation schemes on Sokoban, which underperformed and scored below the standard MCTS.
>
> Thank you for the comment concerning the bias. Besides the "macro-actions interpretation’ mentioned above, our method is a way to deal with the bias-variance problem. The series of papers [Silver, D. et al. 2016, 2017, 2018] provided quite substantial evidence that the MCTS planner with value function evaluation (AlphaZero) replacing the policy rollouts (AlphaGo) is more powerful and simpler. Among multiple reasons, it reduces variance Monte-Carlo estimators provided by the rollouts. Making long rollouts is also more wasteful, exemplified by our random shooting experiments (which achieve decent results but also at a higher computational cost). As such, we are of the opinion that STS is a valuable tool in the current state-of-the-art planning landscape: the idea is simple, easy to code, and can be implemented on top of many algorithms from the MCTS family. Nevertheless, the proposed ablation is interesting. We will run it and update our answer later on.
>
> Concerning the question about pretrained value function for Sokoban experiments presented in Table 1 (and Table 4), we clarify that it came from a separate MCTS based training. More details will be included in the revision.
> Regarding the difference in hyperparameters of STS and Shooting presented in Appendix A.2., we confirm that this is indeed the case. We tuned each of the methods separately to ensure a fair and meaningful comparison (we have run several thousand experiments for each method to obtain the final values of hyperparameters).
>
> Our zero-initialization scheme for the value network is meant to ensure uniform exploration in all directions at the initial training stages. Recent large scale experiments [Andrychowicz M. et al., 2020] suggest that initialization has an important effect on RL training and suggest routine use schemes similar to ‘zero-initialization’, at least in a model-free setting. This is consistent with our observations. We have also experimented with optimistic (as well as pessimistic) initialization, but we have found zero initialization to perform better.
>
> In the PPO shooting experiments, we used the PPO trained policy but not the value function. Therefore the planning was done using as a signal the truncated empirical return gathered on sampled trajectories of length 10. The worse results stem from this shortsightedness. We made this experiment mostly for the benchmarking reasons, as such it will be moved to the appendix in the revision.

---

> ### Author Response · Authors · 2020-11-20
> **We update with new experimental results**
>
> We update with new experimental results. We also point out the reviewer's attention to the new answer to Rev2 in which we show results comparing our methods to AlphaGo.
>
> We conducted experiments on Sokoban with a simpler backpropagation scheme similar to Soemers et al (2016). Namely, we backpropage only the value from the last node of the rollout, instead of the ‘mega-backprop’ of STS. The results are much worse - after 12 million steps STS with such simple backpropagation obtained on average 79% solve ratio (compared to 89% for STS presented in paper). (The graph is available [here](https://postimg.cc/3yztdV0k). Results are averaged over 10 training runs.)

---

### Decision · Program_Chairs · 2021-01-07
**Final Decision**

**Decision:**

Reject

**Comment:**

This paper proposes a modification to MCTS in which a sequence of nodes (obtained by following the policy prior) are added to the search tree per simulation, rather than just a single node. This encourages deeper searches that what is typically attained by vanilla MCTS. STS results in slightly improved performance in Sokoban and much larger improvements Google Research Football.

R4 and R1 both liked the simplicity of the idea, with R1 also praising the paper for the thoroughness of its evaluation. I agree that the idea is interesting and worth exploring, and am impressed by the scope of the experiments in the paper as well as the additional ones linked to in the rebuttal. However, R1 and R5 explicitly noted they had many points of confusion, and across the reviews there seemed to be many questions regarding the difference between STS and other variants of MCTS. I also needed to read parts of the paper multiple times to fully understand the approach. If this many experts on planning and MCTS are confused, then I think readers who are less familiar with the area will definitely struggle to understand the main takeaways. While I do think the clarifications and new experiments provided in the rebuttal help, my overall sense is that the paper at this stage is not written clearly enough to be ready for publication at ICLR. I would encourage the authors to try to synthesize their results and organize them more succinctly in future versions of the paper.

One comment about a point of confusion that I had: I noticed the PUCT exploration parameter was set to zero for Sokoban, and one for GRF (with an explanation given that many values were tried, though these values are unspecified). As the exploration parameter is normally considered to be the thing that controls whether MCTS acts more like BFS ($c = \infty$) or DFS ($c = 0.0$), I would encourage the authors to more explicitly report which values they tried and to be clearer about the advantage of STS's multi-step expansions over low values of the exploration parameter.